# Gonococcal invasion into epithelial cells depends on both cell polarity and ezrin

Qian Yu[1], Liang-Chun Wang[2], Sofia Di Benigno[1], Daniel C. Stein[1], Wenxia Song[1]*

**1** Department of Cell Biology and Molecular Genetics, University of Maryland, College Park, Maryland, United States of America, **2** Marine & Pathogenic Microbiology Lab, National Sun Yat-Sen University, Kaohsiung, Taiwan

* wenxsong@umd.edu

**Data Availability Statement:** All relevant data are within the manuscript and its Supporting Information files.

**Funding:** This work was supported by grants from the National Institutes of Allergy and Infectious

## Abstract

*Neisseria gonorrhoeae* (GC) establishes infection in women from the cervix, lined with heterogeneous epithelial cells from non-polarized stratified at the ectocervix to polarized columnar at the endocervix. We have previously shown that GC differentially colonize and transmigrate across the ecto and endocervical epithelia. However, whether and how GC invade into heterogeneous cervical epithelial cells is unknown. This study examined GC entry of epithelial cells with various properties, using human cervical tissue explant and non-polarized/polarized epithelial cell line models. While adhering to non-polarized and polarized epithelial cells at similar levels, GC invaded into non-polarized more efficiently than polarized epithelial cells. The enhanced GC invasion in non-polarized epithelial cells was associated with increased ezrin phosphorylation, F-actin and ezrin recruitment to GC adherent sites, and the elongation of GC-associated microvilli. Inhibition of ezrin phosphorylation inhibited F-actin and ezrin recruitment and microvilli elongation, leading to a reduction in GC invasion. The reduced GC invasion in polarized epithelial cells was associated with non-muscle myosin II-mediated F-actin disassembly and microvilli denudation at GC adherence sites. Surprisingly, intraepithelial GC were only detected inside epithelial cells shedding from the cervix by immunofluorescence microscopy, but not significantly in the ectocervical and the endocervical regions. We observed similar ezrin and F-actin recruitment in exfoliated cervical epithelial cells but not in those that remained in the ectocervical epithelium, as the luminal layer of ectocervical epithelial cells expressed ten-fold lower levels of ezrin than those beneath. However, GC inoculation induced F-actin reduction and myosin recruitment in the endocervix, similar to what was seen in polarized epithelial cells. Collectively, our results suggest that while GC invade non-polarized epithelial cells through ezrin-driven microvilli elongation, the apical polarization of ezrin and F-actin inhibits GC entry into polarized epithelial cells.

## Author summary

*Neisseria gonorrhoeae* (GC) causes gonorrhea in women by infecting the female reproductive tract. GC entry of epithelial cells has long been observed in patients' biopsies and

Diseases, AI141894 to WS and DCS and AI123340 to DCS and WS. SDB was supported by the National Institutes of Health T32 grant. The funders had no role in study design, data collection and analysis, decision to publish, or preparation of the manuscript.

**Competing interests:** The authors have declared that no competing interests exist.

studied in various types of epithelial cells. However, how GC invade into the heterogeneous epithelia of the human cervix is unknown. This study reveals that both the expression level of ezrin, an actin-membrane linker protein, and the polarization of ezrin-actin networks in epithelial cells regulate GC invasion. GC interactions with non-polarized squamous epithelial cells expressing ezrin induce ezrin activation, ezrin-actin accumulation, and microvilli elongation at GC adherent sites, leading to invasion. Low ezrin expression levels in the luminal ectocervical epithelial cells are associated with low levels of intraepithelial GC. In contrast, apical polarization of ezrin-actin networks in columnar endocervical epithelial cells reduces GC invasion. GC interactions induce myosin activation, which causes disassembly of ezrin-actin networks and microvilli modification at GC adherent sites, extending GC-epithelial contact. Expression of opacity-associated proteins on GC promotes GC invasion by enhancing ezrin-actin accumulation in squamous epithelial cells and inhibiting ezrin-actin disassembly in columnar endocervical epithelial cells. Thus, reduced ezrin expression and ezrin-actin polarization are potential ways for cervical epithelial cells to curtail GC invasion.

## Introduction

Gonorrhea, caused by a gram-negative bacterium *Neisseria gonorrhoeae* (GC), is the second most commonly reportable sexually transmitted infection (STI) in the United States. It has recently become a public health crisis due to the emergence of antibiotic-resistant strains and a lack of vaccines [1–3]. A majority (50%~80%) of female gonococcal infections do not display any symptoms [4,5]. GC can colonize the lower genital tract asymptomatically for a long time, and these asymptomatic infections can lead to severe complications [4]. When GC reach the upper genital tract, the infection can cause pelvic inflammatory disease (PID), a major cause of chronic abdnominal pain, infertility, and predispose women to life-threatening ectopic pregnancy [6]. When entering the bloodstream, the infection can lead to disseminated gonococcal infection (DGI) [6,7]. The asymptomatic nature of female infection allows GC to transmit silently and increases the risk of coinfections with HIV and other STIs [8].

The cervix is the gate from the unsterile lower female reproductive tract to the sterile upper tract [9] and a potential anatomic location that determines whether the outcome of GC infection is asymptomatic or symptomatic. The mucosal surface of the human cervix is characterized by heterogeneous epithelia, multiple-layered non-polarized stratified epithelial cells at the ectocervix and single-layered polarized columnar epithelial cells at the endocervix [10,11]. The heterogeneous cervical epithelial cells build complicated physical and immune barriers against pathogens and also create a technical challenge for generating *in vitro* models that mimic the *in vivo* properties of the cervical surface to understand GC infection mechanism in females. We have established a human cervical tissue explant model [12] and shown that cervical epithelial cells in this explant model maintain most of the *in vivo* properties in culture and that GC infection in this explant model mimics what was observed in patients' biopsies [13,14].

GC establish infection by adhering to, invading into, and/or transmigrating across the epithelium [15–17]. GC express multiple adhesive molecules that can undergo phase and antigenic variation, such as pili, opacity-associated proteins (Opa), and lipooligosaccharide (LOS) [18–21]. These surface molecules mediate GC-epithelial interactions by binding to host cell receptors, establishing colonization. Pili initiate the interaction [22,23], which brings GC close to epithelial cells and enables Opa and LOS build intimate interactions of GC with the plasma membrane of epithelial cells [15,17]. Such interactions lead to GC entry into epithelial cells

[17,24–26]. In polarized epithelial cells, GC invading from the apical surface can transcytose through the cell and exit from the basolateral surface [27–29]. We have previously shown that GC interaction with polarized epithelial cells induces the disassemble of the apical junction, which allows GC transmigration across the epithelium [14]. GC-induced epithelial junction disassembling requires $Ca^{2+}$-dependent activation of non-muscle myosin II (NMII), a part of the actomyosin supporting structure for the apical junction [14]. Using the human cervical tissue explant model, we have shown that GC differentially infect the various regions of the cervix when the different regions are equally exposed to the bacteria. GC preferentially colonize the ectocervical epithelium, using an integrin β1-dependent mechanism, and exclusively penetrate into the subepithelium of the endocervix by disrupting the apical junction using the NMII-dependent mechanism [13]. Pili are essential for GC colonization and penetration in all cervical regions. Expression of Opa proteins that bind carcinoembryonic antigen-related cell adhesion molecules (CEACAMs) as host receptors enhances colonization but reduces GC penetration [13]. However, whether and how GC invade into various cervical epithelial cells remains unknown.

Intracellular GC were first observed in epithelial cells of the fallopian tube [30,31]. GC invasion into epithelial cells has been since studied extensively using various GC strains and epithelial cell lines [17,21,32–36], as well as primary human cervical epithelial cells [25,37]. While most previous studies were based on non-polarized epithelial cells, GC entry into polarized epithelial cells has been reported in human colonic epithelial cells T84 [28,33] and columnar epithelial cells of the fallopian tube [30,31]. Pili, Opa, LOS, and porin all have been shown to be involved in GC invasion, but the level of their involvement depends on the types of epithelial cells and GC variants [4,13,26,38–40]. Importantly, GC entry various epithelial cells all requires actin reorganization [24,26,34,41]. This reorganization is triggered by signaling induced by the binding of GC pili, Opa, LOS, and porin to host cell receptors, such as pili binding of complement receptor 3 and Opa binding of CEACAMs and heparan sulfate proteoglycans (HSPG) [26,34,42–45]. F-actin and ezrin concentrate at GC adherent sites, supporting microvilli elongation and membrane ruffling around bacteria, allowing or promoting for invasion [17,24,44–46]. However, whether GC utilize a similar mechanism to enter human ectocervical and endocervical epithelial cells in intact tissues is unknown.

Polarity is one of the properties that distinguish the ecto and endocervical epithelial cells. Ectocervical epithelial cells are flat and stratified with relatively smooth surfaces and low levels of cell polarity. In contrast, endocervical epithelial cells are tall and columnar with densely packed microvilli at the apical surface and high levels of cell polarity [10,47]. Polarized columnar endocervical epithelial cells form the apical junction complexes with neighboring cells, which hold columnar epithelial cells together into a monolayer, seal spaces between cells to build a physical barrier, and facilitate the polarized distribution of surface molecules [48]. The actin cytoskeleton in ectocervical epithelial cells shows no polarity, distributing evenly at the cell periphery, supporting the cell membrane. In contrast, endocervical epithelial cells exhibit strong polarity of the actin cytoskeleton, concentrating under the apical membrane and supporting the apical junction and dense microvilli [49]. Actin-mediated morphological changes in both non-polarized and polarized epithelial cells, including microvilli formation, have been shown to require ezrin [50,51]. Upon phosphorylation, ezrin interacts with both F-actin and membrane proteins, linking the actin cytoskeleton to the plasma membrane of epithelial cells [52]. Ezrin has also been suggested to play a role in microbial infection of epithelial cells, including GC [44,53], *Neisseria meningitis* [54,55], *Chlamydia trachomatis* [56], *Group A Streptococcus* [57], *Enteropathogenic Escherichia coli* [58], and *Pseudomonas aeruginosa* [59]. Whether differences in epithelial cell polarity and morphology and the underlying actin organization impact GC infection at various cervical regions is unclear.

This study examined the effect of polarity, morphology, and actin organization in ecto and endocervical epithelial cells on GC invasion and the underlying mechanism, using a non-polarized and polarized epithelial cell line model generated from the same cell line and the human cervical tissue explant model. We found that the polarity of epithelial cells and the expression of ezrin are two factors regulating the efficiency of GC invasion into epithelial cells. We further investigated the mechanisms by which the polarity of cervical epithelial cells and the level of ezrin expression regulate GC invasion.

## Results

### GC invade into non-polarized epithelial cells more efficiently than polarized epithelial cells

To determine if cell polarity alone affects GC infectivity, we established non-polarized and polarized epithelial cells on transwells with the same numbers of human colonic epithelial cell line T84, using a previously published protocol with modification [32,33]. After comparing multiple types of human epithelial cells, including endometrial epithelial cells HEC-1-B, primary cervical epithelial cells [45], and cervical epithelial cells immortalized by HPV [60], we chose T84, as it has been used to define how GC interact with polarized epithelial cells [27,28,33,61–63] and it was the only cell line that can be polarized to the level and exhibit the columnar morphology, the barrier function, and apically polarized distribution of CEACAMs [64], similar to endocervical epithelial cells *in vivo* (S1 and S2 Figs) [13,14,33,64]. The two-day culture of T84 cells on transwells was modeled as non-polarized cells, as monolayers had low transepithelial resistance (TEER) (<500 Ω), were permeable to the CellMask dye (S1 Fig), and showed flat morphology and random distribution of F-actin and the junction proteins E-cadherin and ZO-1 at the cell periphery (S2 Fig). After culturing on transwells for ~10 days, T84 cells became polarized, exhibiting > 2000 Ω TEER, the barrier function against the dye (S1 Fig), and apical polarization of F-actin, E-cadherin, and ZO-1 staining (S2 Fig).

Using the cell line model, we examined the effect of epithelial morphology and polarity on the adhesion and invasion of piliated WT MS11 GC that expressed phase variable Opa (Opa+) using the gentamicin resistance assay [26,32,65]. GC were inoculated from the top of transwell chambers at an MOI of 1:10 and incubated for 3, 6, and 12 h. After washing, epithelial cells were either lysed to release all epithelial-associated bacteria or treated with 100 μg/ml gentamicin for 2 h before lysis to quantify intracellular bacteria. After 3-h inoculation, there were $>1 \times 10^6$ associating with T84 cells and $\geq 1 \times 10^3$ gentamicin-resistant bacteria in each transwell (Fig 1A and 1B). An additional 3-h incubation slightly increased the numbers of gentamicin-resistant but not epithelial-associated GC (Fig 1A and 1B). By 12 h, the numbers of epithelial-associated and gentamicin-resistant GC both reduced due to overgrowth of bacteria, even when we removed unassociated GC at 6 h (Fig 1A and 1B). Thus, we choose 6 h time points for determining epithelial-associated and invaded GC, respectively. Notably, the numbers of gentamicin-resistant GC were significantly higher in non-polarized than polarized T84 cells at both 3 and 6 h (Fig 1B), but the numbers of GC associating non-polarized and polarized T84 cells were similar (Fig 1A). Consequently, the percentage of intracellular GC among the total epithelial-associated GC in non-polarized T84 cells was significantly higher than that in polarized T84 cells (Fig 1C). These results suggest that the polarization of epithelial cells reduces GC invasion.

As Opa proteins are known to be involved in both GC adherence and invasion [41,66,67], we determined whether Opa proteins contributed to the difference in GC invasion into non-polarized and polarized epithelial cells using piliated WT MS11 (Opa+), MS11 with all *opa* genes deleted (ΔOpa), and isogenic derivative of MS11 expressing CEACAM-binding OpaH

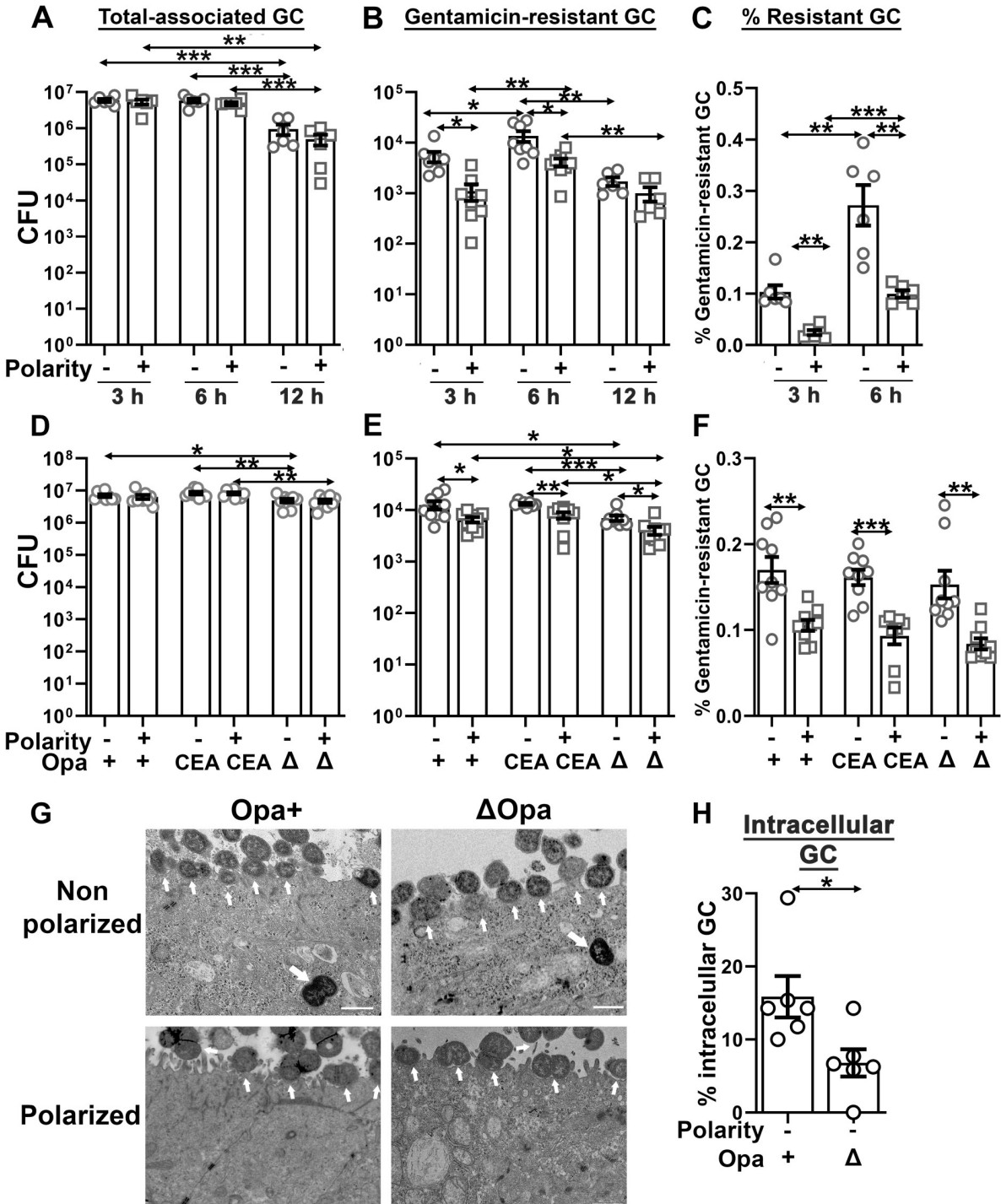

**Fig 1. The polarity of epithelial cells and Opa expression regulate GC infection.** (A-C) Non-polarized and polarized T84 cells were inoculated with Pil+Opa+ GC (MOI~10) from the top chamber for 3, 6, and 12 h. Infected epithelial cells were lysed before and after gentamicin treatment to determine total epithelial-associated GC (±SEM) **(A)** and gentamicin-resistant GC (±SEM) **(B)**. The percentage of gentamicin-resistant GC among total epithelial-associated GC were calculate **(C)**. (D-F) Cells were inoculated with Pil+Opa+, Opa$_{CEA,}$ or ΔOpa GC (MOI~10) from the top chamber for 6 h. Total epithelial-associated GC (±SEM) **(D)**, gentamicin-resistant GC (±SEM) **(E)**, and the percentage of gentamicin-resistant GC among total epithelial-associated GC were quantified **(F)**. Data points represent individual transwells **(A-F)**. **(G, H)** Cells were incubated with Pil+Opa+ or Pil+ΔOpa GC (MOI~50) from the top chamber for 6 h, fixed, and processed for transmission electron microscopy (TEM). Representative images of GC-inoculated non-polarized and polarized T84 cells are shown **(G)**. Small arrows, GC directly contacting epithelial membranes; and big arrow, intracellular GC. The level of GC invasion in non-

polarized T84 was estimated by percentages of the intracellular GC among the total number of GC directly contacting epithelial cells in each of images showing intracellular GC (**H**). Data points represent individual images. Scale bar, 1 μm. n = 3~4 three to four independent experiments and two to three transwells per experiment. *$p<0.05$; **$p<0.01$; ***$p<0.001$.

(Opa$_{CEA}$) that cannot undergo phase variation [68]. When comparing different GC variants, we found that the numbers of both epithelial-associated (Fig 1D) and gentamicin-resistant Opa+ and Opa$_{CEA}$ GC (Fig 1E) were significantly higher than those of ΔOpa GC in both polarized and non-polarized T84 cells after 6-h incubation, consistent with what was previously reported [41,66,67]. However, the percentages of gentamicin-resistant GC relative to the total epithelial-associated GC were similar among the three variants (Fig 1F). We next examined Opa's role in GC adherence and invasion using an anatomically relevant cell line, ME-180. Similarly, the numbers of both epithelial-associated (S3A Fig) and gentamicin-resistant Opa + and Opa$_{CEA}$ GC (S3B Fig) were higher than those of ΔOpa GC, confirming what we observed with non-polarized T84 cells (Fig 1D and 1E). When comparing GC adherence and invasion in non-polarized and polarized T84 cells, we found that each of the three MS11 variants attached to non-polarized and polarized T84 cells at similar levels (Fig 1D), but their gentamicin resistant numbers (Fig 1E) and percentages (Fig 1F) were all higher in non-polarized than polarized T84. These results suggest that Opa expression enhances both GC adherence and invasion into non-polarized and polarized epithelial cells, and the polarization of epithelial cells only reduces GC invasion without affecting GC adherence.

We further examined the effects of cell polarity and Opa expression on GC entry into epithelial cells using transmission electron microscopy (TEM). Non-polarized and polarized T84 cells were incubated with Opa+ and ΔOpa at an MOI of 1:50 for 6 h. Average eight images were randomly acquired from each condition and each of three independent experiments. While Opa+ and ΔOpa GC were able to form tight associations with both non-polarized and polarized T84 cells (Fig 1G), we were only able to visualize intracellular GC in ~25% of the images from GC-inoculated non-polarized T84 cells but none of the eight images from GC-inoculated polarized T84 cells. Our inability to detect intracellular GC in polarized T84 cells was surprising, probably due to a limited number of images acquired. Using images with intracellular GC, we estimated the percentage of intracellular GC among the total number of GC directly contacting epithelial cells in individual images (Figs 1G, 1H, and S4; big arrows, intracellular GC and small arrows, GC directly contacting the plasma membrane of epithelial cells). We found a significantly higher percentage of intracellular Opa+ (~16%) than ΔOpa GC (~7%) in non-polarized T84 cells (Fig 1H). Hence, the results of TEM analysis support an enhancing effect of Opa and a suppressing effect of epithelial polarity on GC invasion.

## GC inoculation induces distinct F-actin reorganization in non-polarized and polarized epithelial cells

The actin cytoskeleton is required for GC invasion [24,26,34,41], and non-polarized and polarized epithelial cells organize actin differently (S2 Fig). We postulated that the difference in actin organization in non-polarized and polarized epithelial cells is one of the underlying causes of differential GC invasion efficiencies. To test this hypothesis, we inoculated non-polarized and polarized T84 cells with piliated Opa+, Opa$_{CEA}$, and ΔOpa GC for 6 h, fixed, stained for F-actin with phalloidin and GC with an polyclonal antibody, and analyzed by three-dimensional confocal fluorescence microscopy (3D-CFM) to examine cells from the top (xy view, Fig 2A, top panels) and the side (xz view, Fig 2A, bottom panels) of the epithelium. We observed different actin reorganization beneath GC adherent sites in non-polarized and polarized epithelial cells (Fig 2A, arrows). We quantified the actin reorganization by the mean

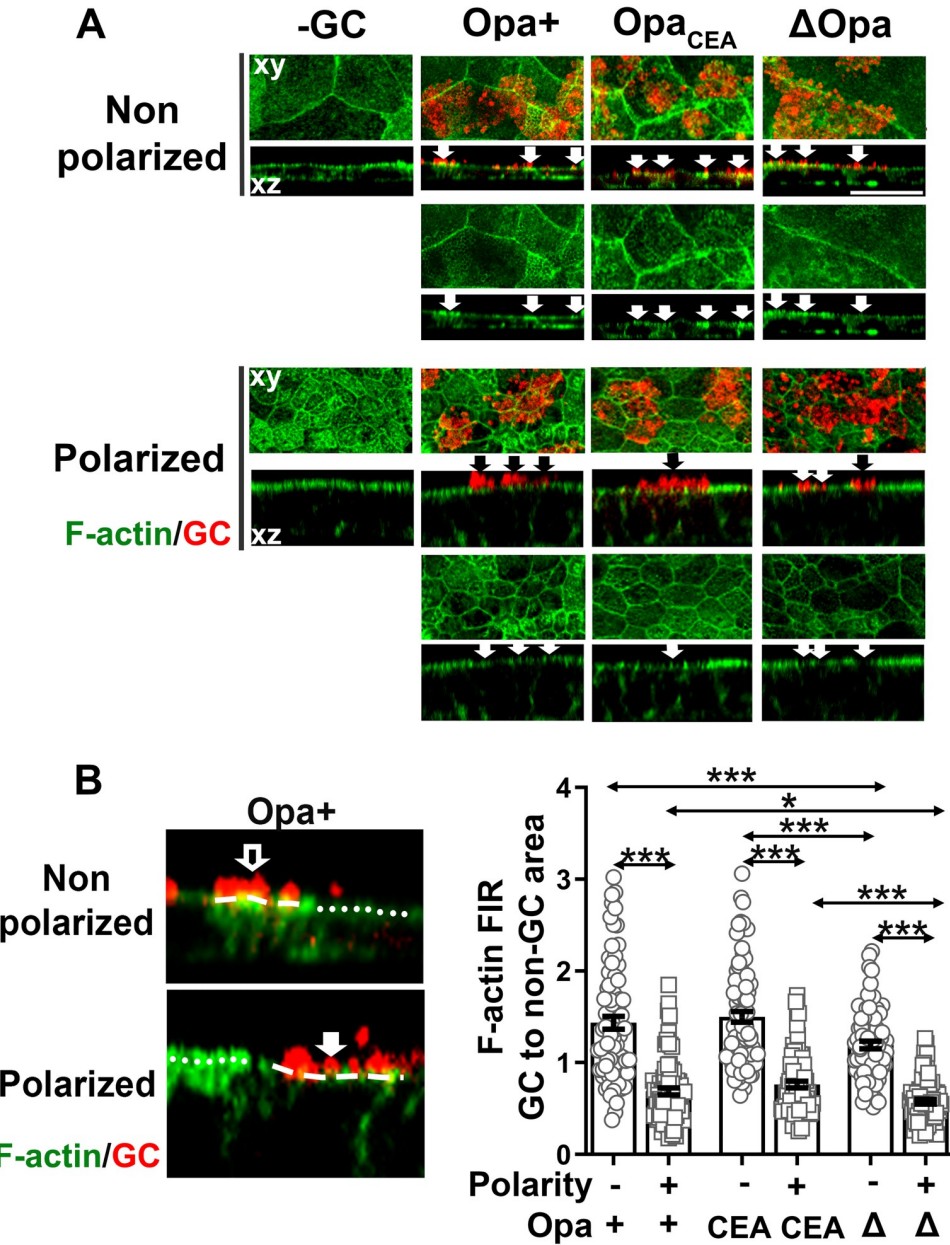

**Fig 2. GC inoculation induces distinct F-actin reorganization in non-polarized and polarized epithelial cells.** Non-polarized and polarized T84 cells were apically inoculated with Pil+Opa+, Opa_CEA or ΔOpa GC (MOI~10) from the top chamber for 6 h. Cells were fixed, stained for F-actin and GC, and analyzed using 3D-CFM. **(A)** Representative xy images of the top surface and xz images crossing the top and the bottom surfaces are shown. Arrows indicate the location of GC. **(B)** The redistribution of F-actin was quantified by the mean fluorescence intensity ratio (FIR) (±SEM) of F-actin underneath individual GC microcolonies relative to the adjacent no GC surface area. Data points represent individual GC microcolonies. Scale bar, 20 μm. n = 3 three independent experiments. $^*p<0.05$; $^{**}p<0.01$; $^{***}p<0.001$.

fluorescent intensity ratio (FIR) of F-actin staining underneath GC microcolonies relative to the adjacent no GC surface area. The FIRs of F-actin staining underneath microcolonies of all three GC variants were higher than 1 in non-polarized epithelial cells (Fig 2B), indicating a localized enrichment of F-actin at GC adherent sites in non-polarized T84 cells. In contrast, the F-actin FIRs were lower than 1 in polarized epithelial cells inoculated with each of the

three GC variants (Fig 2B), indicating a localized reduction of F-actin at GC adherent sites in polarized T84 cells. When comparing the three GC variants, the F-actin FIRs underneath Opa+ and Opa$_{CEA}$ GC were significantly higher than those underneath ΔOpa GC in both non-polarized and polarized epithelial cells (Fig 2B). We also observed a similar F-actin enrichment at GC adherent sites in ME-180 cells cultured on transwells for 2 days (S5A Fig). The F-actin FIRs underneath Opa+ and Opa$_{CEA}$ were also significantly higher than those underneath ΔOpa GC (S5B Fig). These data suggest that GC can induce opposite actin reorganization in non-polarized (F-actin enrichment) and polarized epithelial cells (F-actin reduction) at adherent sites. Opa expression enhances the F-actin enrichment in non-polarized epithelial cells and alleviates the F-actin reduction in polarized epithelial cells.

## GC inoculation activates and recruits ezrin to adherent sites in non-polarized epithelial cells, facilitating F-actin enrichment and GC invasion

Ezrin functions as a linker between the actin cytoskeleton and the plasma membrane, and its linker function is activated through phosphorylation of threonine 567 [52]. We determined whether GC induced differential actin reorganization in non-polarized and polarized epithelial cells through ezrin by analyzing ezrin cellular distribution and phosphorylation in response to GC inoculation. We inoculated non-polarized and polarized T84 cells with piliated Opa+, Opa$_{CEA}$, and ΔOpa GC for 6 h. Cells were fixed, stained for ezrin and GC, and analyzed by 3D-CFM. We evaluated the redistribution of ezrin staining underneath GC adherent sites relative to the adjacent no GC surface area by FIR. In the absence of GC, the ezrin staining was distributed randomly in the cytoplasm and periphery of non-polarized epithelial cells, but concentrated at the apical surface in polarized epithelial cells (Fig 3A), similar to the distribution of F-actin (Fig 2A). Inoculation of each of three GC variants all induced ezrin enrichment at adherent sites of non-polarized epithelial cells (Fig 3A, upper panels, arrows), leading to ezrin FIRs > 1 (Fig 3B). In contrast, we observed no change or a reduction in ezrin staining at Opa+, Opa$_{CEA}$, and ΔOpa GC adherent sites at the apical surface of polarized epithelial cells (Fig 3A, bottom panels, arrows). Consequently, their ezrin FIRs were ≤ 1 and were significantly lower than those in non-polarized epithelial cells (Fig 3B). Furthermore, the ezrin FIRs underneath Opa+ and Opa$_{CEA}$ GC were significantly higher than ΔOpa GC in both polarized and non-polarized T84 cells (Fig 3B). We also observed the recruitment of ezrin to GC adherent sits in ME180 cells (S5C Fig) and higher ezrin FIRs underneath Opa+ and Opa$_{CEA}$ than ΔOpa GC (S5D Fig). These data suggest that GC inoculation causes ezrin recruitment and disassociation at adherent sites in non-polarized and polarized epithelial cells, respectively, similar to GC-induced actin reorganization. Opa expression enhances the ezrin recruitment in non-polarized epithelial cells but alleviates the ezrin disassociation in polarized epithelial cells.

GC-induced ezrin redistribution suggests a role for GC in regulating ezrin activation. To test this, we determined the levels of ezrin phosphorylation at T567 (pT567) using western blotting 6 h post-incubation (Fig 3C). Using the density ratio of ezrin pT567 and total ezrin, we found a significant increase in ezrin phosphorylation in GC inoculated non-polarized epithelial cells compared to no GC controls (Fig 3C and 3D). However, GC inoculation did not significantly change the phosphorylation level of ezrin in polarized epithelial cells. We did not detect any significant difference in ezrin phosphorylation levels between Opa+ and ΔOpa GC-inoculated epithelial cells, no matter if they were polarized or not. There was also no significant difference in the density ratio of ezrin to β-tubulin in response to GC inoculation in both non-polarized and polarized epithelial cells (Fig 3C and 3E). Thus, GC inoculation induces ezrin

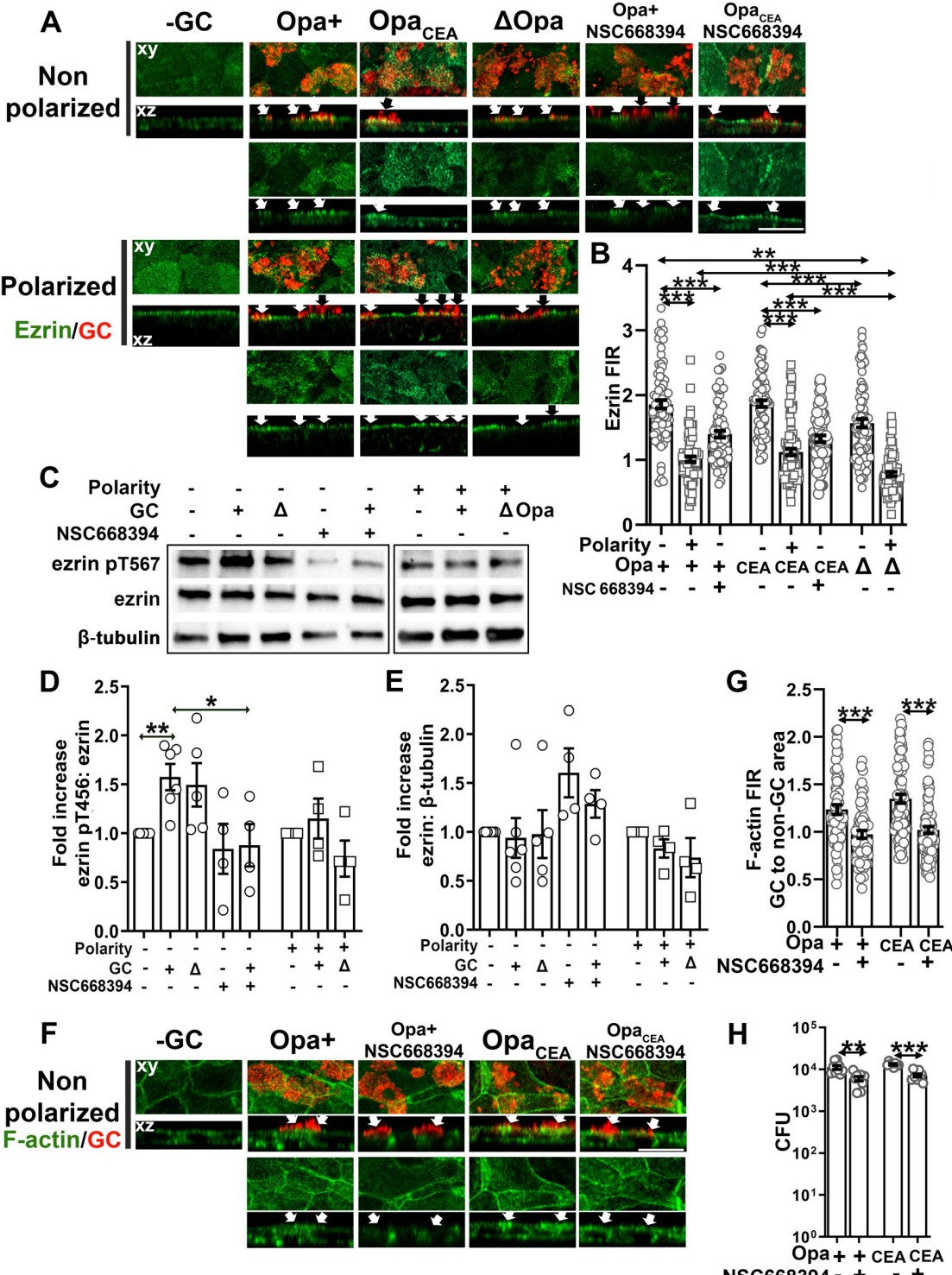

**Fig 3. GC induce differential activation and redistribution of ezrin in non-polarized and polarized epithelial cells for actin reorganization and GC invasion.** Non-polarized and polarized T84 were pretreated with or without the ezrin activation inhibitor NSC668394 (20 μM) for 1 h and apically inoculated with Pil+Opa+, Opa$_{CEA}$, or ΔOpa GC (MOI~10) for 6 h with or without the inhibitor. **(A, B, F, G)** Cells were fixed, permeabilized, stained for ezrin **(A)**, F-actin **(F)**, and GC, and analyzed using 3D-CFM. Shown are representative xy and xz images **(A, F)**. The levels of ezrin **(B)** and F-actin **(G)** redistribution were quantified by FIR (±SEM) underneath GC microcolonies relatively to the adjacent no GC surface area. Data points represent individual GC microcolonies. **(C-E)** Cells were lysed and analyzed by western blot. Shown are representative blots **(C)**. The average fold of

increase in the ezrin pT567 to ezrin ratio **(D)** and the ezrin to β-tubulin ratio **(E)** (±SEM) was quantified by NIH ImageJ. Data points represent individual transwells. **(H)** Invaded GC (±SEM) were quantified by gentamicin resistance assay 6 h post-inoculation with or without NSC668394 treatment. Data points represent individual transwells. Scale bar, 20 μm. n = 2~3 two to three independent experiments. $^*p<0.05$; $^{**}p<0.01$; $^{***}p<0.001$.

activation exclusively in non-polarized epithelial cells. The above data suggest that GC induce ezrin activation in non-polarized epithelial cells but not in polarized epithelial cells.

The activation and recruitment of ezrin to GC adherent sites of non-polarized T84 cells suggest a role for ezrin in GC-induced actin reorganization and GC infection. We examined this hypothesis utilizing an ezrin inhibitor NSC668394, which blocks ezrin phosphorylation at threonine 567 [69]. We determined the effect of the inhibitor on ezrin and actin reorganization by immunofluorescence microscopy and FIR as described above and on GC invasion by the gentamicin resistance assay. The ezrin inhibitor significantly reduced the FIRs of ezrin underneath the adherent sites of both Opa+ and Opa$_{CEA}$ GC relative to the no GC area (Fig 3A and 3B) and ezrin phosphorylation without affecting the protein level of ezrin (Fig 3C–3E), indicating effective inhibition of ezrin activation. Importantly, the inhibitor treatment significantly reduced the FIRs of F-actin underneath the adherent sites of both Opa+ and Opa$_{CEA}$ GC relative to the no-GC area (Fig 3F and 3G) as well as the numbers of gentamicin-resistant Opa + and Opa$_{CEA}$ GC (Fig 3H), compared to no inhibitor controls. Similarly, the ezrin inhibitor also reduced the FIRs of ezrin and F-actin underneath the adherent sites of Opa$_{CEA}$ GC relative to the no-GC area (S5 Fig) and the number of gentamicin-resistant Opa$_{CEA}$ GC (S3 Fig) in ME180 cells. However, the ezrin inhibitor had no significant effect on GC growth (S6A Fig) and the total number of GC associating non-polarized T84 (S6B Fig). These data suggest that the activation and recruitment of ezrin to GC adherent sites in non-polarized epithelial cells leads to F-actin accumulation, facilitating GC invasion.

## GC induce the activation and recruitment of NMII to GC adherent sites in polarized epithelial cells, causing actin depolymerization

We have previously shown that GC inoculation induces NMII activation and recruitment to GC adherent sites in polarized epithelial cells in a calcium-dependent manner, leading to apical junction disassembly and gonococcal transmigration across the epithelium [14]. Here, we addressed whether NMII, an actin motor, was involved in the differential F-actin remodeling in non-polarized and polarized epithelial cells. We first determined whether GC could induce NMII activation and redistribution in non-polarized epithelial cells, similarly to what we observed in polarized epithelial cells, using immunofluorescence microscopy. In the absence of GC, phosphorylated NMII light chain (pMLC) staining was distributed randomly at the cell periphery and the cell-cell contact in non-polarized epithelial cells but concentrated at the apical part of polarized epithelial cells (Fig 4A). We did not observe significantly changes in pMLC distribution of non-polarized epithelial cells inoculated with Pil+Opa+ or Pil+ΔOpa GC (Fig 4A, top panels; and arrows, GC microcolonies), which is supported by the FIR of pMLC staining at GC adherent sites compared to no GC surface area (~1) (Fig 4B). This was in sharp contrast with what observed in polarized epithelial cells–pMLC concentrated at GC adherent sites at the apical surfaces (Fig 4A, bottom panels, arrows), resulting in the FIRs of pMLC at GC adherent sites relative to no GC surface area higher than 1 (Fig 4B). Pil+ΔOpa-infected polarized epithelial cells exhibited significantly higher pMLC FIRs (~ 1.9) than Pil +Opa+ GC-infected polarized epithelial cells (~ 1.6) (Fig 4B). These data suggest that GC inoculation only induces the activation and recruitment of NMII in the polarized but not non-polarized epithelial cells, and Opa expression suppresses such effects.

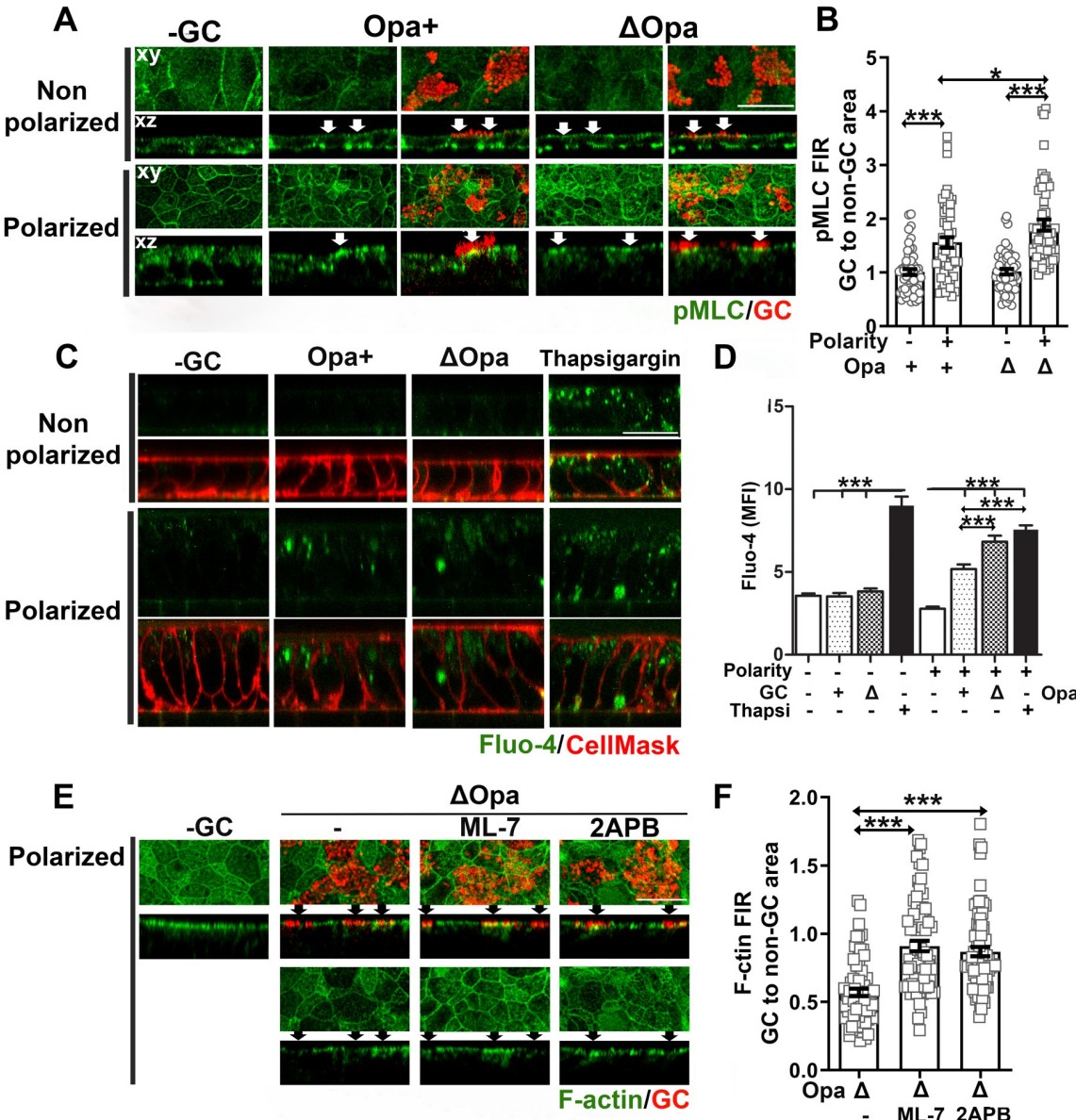

**Fig 4. NMII activation and Ca²⁺ elevation are required for GC-induced F-actin reduction at adherent sites of polarized epithelial cells. (A, B)** Non-polarized and polarized T84 cells were inoculated with Pil+Opa+ or ΔOpa GC from the top chamber for 6 h (MOI~10). Cells were fixed and stained for phosphorylated myosin light chain (pMLC) and GC and analyzed using 3D-CFM. Shown are representative xy images of the top surface, xz images crossing both the top and the bottom surface **(A),** and the FIR (±SEM) of pMLC staining underneath individual GC microcolonies relative to the adjacent no GC surface area **(B)**. **(C, D)** Non-polarized and polarized T84 cells were incubated from the top chamber with or without GC Pil+Opa+ or Pil+ΔOpa in the presence or absence of thapsigargin (10 µM) for 4 h. Cells were incubated with the Ca²⁺ indicator Fluo4 and the membrane dye CellMask and analyzed using 3D-CFM. Shown are representative xz images **(C)** and the mean fluorescence intensity (MFI) (±SEM) of Fluo-4 in the cytoplasmic region of individual epithelial cells **(D)**. **(E, F)** Polarized T84 cells were pre-treated with or without ML-7 (10 µM) or 2APB (10 µM) and apically inoculated with Pil+ΔOpa GC in the presence or absence of inhibitors for 6 h (MOI~10). Cells were fixed and stained for F-actin using phalloidin and GC using antibody and analyzed using 3D-CFM. Shown are representative xy and xz images **(E)** and the FIR (±SEM) of phalloidin staining underneath individual GC microcolonies relative to the adjacent no GC surface area **(F)**. Data points in (B) and (F) represent individual GC microcolonies. Scale bar, 20 µm. n = 3 three independent experiments. *$p<0.05$; ***$p<0.001$.

We next determined whether the absence of the NMII activation and recruitment in non-polarized epithelial cells is due to a lack of $Ca^{2+}$ flux response exhibited by polarized epithelial cells [14]. We measured the cytoplasmic $Ca^{2+}$ level at 4 h post GC inoculation, using Fluo-4 AM ester, a calcium dye, and 3D-CFM. Thapsigargin treatment, which elevates intracellular $Ca^{2+}$ [70], served as a positive control. Consistent with previously published studies [14], GC inoculation increased the intracellular $Ca^{2+}$ level of polarized epithelial cells, with the $Ca^{2+}$ level in Pil+ΔOpa GC-infected cells significantly higher than Pil+Opa+ GC-infected cells (Fig 4C and 4D). However, GC inoculation did not change $Ca^{2+}$ levels in non-polarized epithelial cells (Fig 4C and 4D). This result suggests that GC inoculation elevates the cytoplasmic $Ca^{2+}$ levels in polarized epithelial cells but not in non-polarized epithelial cells.

To determine whether GC-induced activation of NMII and $Ca^{2+}$ flux contributed to GC-induced actin reorganization in polarized epithelial cells, we utilized an MLC kinase (MLCK) inhibitor, ML-7 [71], and a $Ca^{2+}$ flux inhibitor, 2APB [72], which have been shown to effectively inhibit GC-induced activation of NMII and $Ca^{2+}$ flux [14]. We evaluated their effects on the F-actin reduction at adherent sites of Pil+Opa+ GC in polarized epithelial cells, using 3D-CFM and the FIR of F-actin at GC adherent sites relative to the no GC surface area (Fig 4E and 4F). Even though each inhibitor have its own off target effects [73,74], both the MLCK and $Ca^{2+}$ inhibitors restored the level of F-actin at GC adherent sites and significantly increased the F-actin FIR from ~0.6 to 0.9 (Fig 4E and 4F). These data together suggest that GC reduce F-actin at adherent sites in polarized epithelial cells by activating $Ca^{2+}$ flux and NMII locally.

## GC differentially remodel microvilli of non-polarized and polarized epithelial cells depending on ezrin- and NMII, respectively

Membrane ruffling, microvilli elongation and denudation have been observed in inoculated primary cervical and endometrial epithelial cells [45,75] and cell lines [17,24,26,76], as well as patient biopsies [77]. Such morphological changes have been implicated in GC invasion. We determined whether the differential actin reorganization we observed causes different morphological changes in non-polarized and polarized T84 cells using TEM. Without GC inoculation, there were short irregular membrane protrusions randomly distributing on the top surface of non-polarized epithelial cells (Figs 5A and S7A, top panels). With GC inoculation, the epithelial membrane contacting GC showed elongated microvilli wrapping around bacteria (Figs 5A and S7A, arrows). We evaluated these morphological changes by measuring the percentage of GC contacting elongated microvilli relative to the total epithelial contacting GC in individual TEM images. We found that ~47% Pil+Opa+ GC and ~33% of Pil+ΔOpa GC with elongated microvilli (Fig 5B). Differing from non-polarized cells, polarized T84 cells have densely packed vertical microvilli at the apical surface in the absence of GC (Figs 5C and S7B, top panels). Post inoculation, the microvilli contacting Pil+Opa+ GC became shorter and bent (Figs 5C and S7B, open arrows). The epithelial membrane contacting Pil+ΔOpa GC lost most microvilli and closely interacted with GC (Figs 5C and S7B, open arrowheads). Quantitative analysis of TEM images showed that microvilli disappeared underneath ~63% Pil+ΔOpa and ~13% Pil+Opa+ GC (Fig 5D). These results suggest that GC induce microvilli elongation in non-polarized epithelial cells but denude microvilli at the contact membrane of polarized epithelial cells. Opa expression promotes microvilli elongation in non-polarized epithelial cells while protecting microvilli in polarized epithelial cells.

We next determined whether GC-induced differential actin reorganization was responsible for the epithelial morphological changes, using the ezrin activation inhibitor NSC668394 and the MLCK inhibitor ML-7, which inhibit GC-induced actin reorganization (Figs 3F, 3G, 4E and 4F). Treatment of NSC668394 significantly reduced the percentage of Pil+Opa+ GC

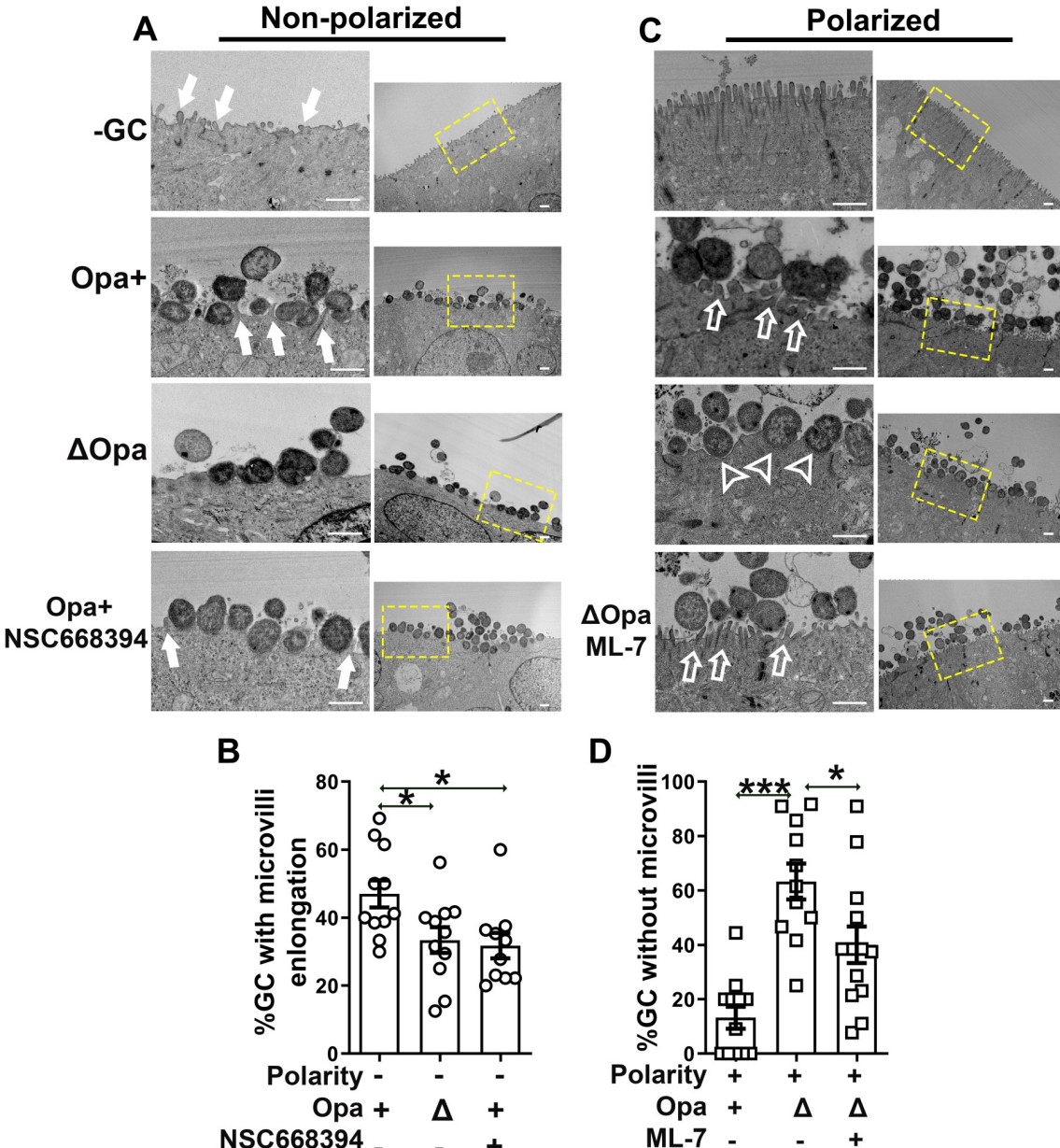

**Fig 5. GC differentially remodel the surface morphology of non-polarized and polarized epithelial cells in an ezrin- or NMII-dependent manner.** Non-polarized and polarized T84 cells were pre-treated with or without NSC668394 (20 μM) or ML-7 (10 μM) and incubated with Pil+Opa+ or Pil+ΔOpa GC (MOI~50) in the apical chamber for 6 h with or without the inhibitors. Cells were fixed and processed for TEM. **(A, C)** Shown are representative images with high (left panels) and low (right panels) magnifications. Yellow dash line rectangles highlight the focused area. Solid arrows, GC-associated elongated microvilli. Open arrows, GC contacting bending microvilli. Open arrowheads, no microvilli visible in GC contact sites. **(B, D)** Membrane morphological changes were quantified by the percentage of GC associating with elongated microvilli in non-polarized epithelial cells **(B)** and the percentage of GC associating with the plasma membrane of polarized epithelial cells that lost microvilli **(D)**. Data points represent individual randomly acquired TEM images. Scale bar, 1 μM. n = 2–3 two to three independent experiments. *$p<0.05$; ***$p<0.001$.

associating with elongated microvilli from ~47% to ~32% in non-polarized epithelial cells (Figs 5A, 5B and S7A). Conversely, treatment of ML-7 significantly reduced the percentage of GC associating with the epithelial membrane that lost microvilli from ~63% to ~40% in polarized epithelial cells (Figs 5C, 5D and S7B). These data suggest that GC adherence induces

actin-driven morphological changes in epithelial cells: localized microvilli elongation in non-polarized epithelial cells by activating ezrin to promote invasion and localized microvilli shortening, tilting, or denudation at the apical surface of polarized epithelial cells by activating NMII and NMII-driven actin depolymerization to facilitate GC-epithelial contact. Opa expression promotes ezrin- and actin-driven microvilli elongation in non-polarized epithelial cells but inhibits NMII-driven microvilli modification in polarized epithelial cells.

## Intraepithelial GC were undetectable in the ectocervix and the endocervix

We evaluated whether GC can invade into the epithelial cells in the human cervix. Human cervical tissue explants were incubated with Pil+Opa+ and Pil+ΔOpa GC (MOI~10) for 24 h, washed at 6 and 12 h post-inoculation to remove unassociated bacteria, and cryopreserved. Tissue sections were stained for GC, F-actin and DNA, and analyzed using CFM as previously described [13]. GC staining was predominantly detected at the luminal surface of both stratified ectocervical and columnar endocervical epithelial cells, occasionally in the subepithelium of the endocervix, but hardly any within the epithelium (Fig 6A). We evaluated GC distribution in cervical tissue explants by measuring the percentage of GC fluorescence intensity (% GC FI) at the luminal surface, within the cervical epithelium, and at the subepithelium. Nearly 100% of GC staining was at the luminal surface of the ectocervical epithelium. No significant GC staining was detected within the ectocervical epithelium and at the subepithelium, no matter if Opa+ or ΔOpa GC was inoculated (Fig 6B). While significant percentages of GC staining were detected at the endocervical subepithelium, the percentage of GC staining within the endocervical epithelium remained very low (Fig 6B). Consistent with our previously published data [13], the percentage of Pil+ΔOpa GC staining at the endocervical subepithelium was significantly higher than that of Pil+Opa+ GC (Fig 6B). While distinguishing intracellular and extracellular GC staining in tissue explants was technically difficult, we noticed GC staining inside epithelial cells shedding from cervical tissues (Figs 6D and S8A, open arrows). However, shedding cervical cells were rarely caught as they were likely disassociated from tissues. Consequently, quantification became impossible.

As distinct actin reorganization leads to different GC invasion levels in non-polarized and polarized epithelial cells, we examined the distribution of F-actin, stained by phalloidin, in cervical tissues in relation to GC locations, using the FIR of phalloidin staining at GC adherent sites relatively to an adjacent luminal area without GC. Contrary to the results from the non-polarized epithelial cell line model shown above, the FIR of F-actin remained at ~1 in stratified non-polarized ectocervical epithelial cells, indicating no F-actin accumulation at GC adherent sites (Fig 6A and 6C). Notably, we observed F-actin accumulation at GC adherent sites of shedding cervical epithelial cells (Figs 6D and S8B, arrows). Similar to what we observed in polarized T84 cells, the F-actin FIR in GC colonized columnar polarized endocervical epithelial cells was reduced below 1 (Fig 6A and 6C), showing F-actin reduction at GC adherent sites. The F-actin FIR in Pil+ΔOpa GC-colonized was significantly lower than that in Pil+Opa+ GC-colonized endocervical epithelial cells (Fig 6C). These data suggest that the number of GC invading into cervical epithelial cells is less than the numbers of GC colonizing the ecto- and endocervix and penetrating the endocervix, which may be associated with GC failure to induce F-actin accumulation at adherent sites. However, GC appear to be able to induce F-actin accumulation at GC adherent sites and enter epithelial cells shedding off the cervix.

## GC inoculation does not affect the organization of ezrin and NMII in ectocervical epithelial cells

To understand our surprising findings of undetectable levels of intraepithelial GC and F-actin accumulation at GC adherent sites in the ectocervix, we examined ezrin and NMII responses

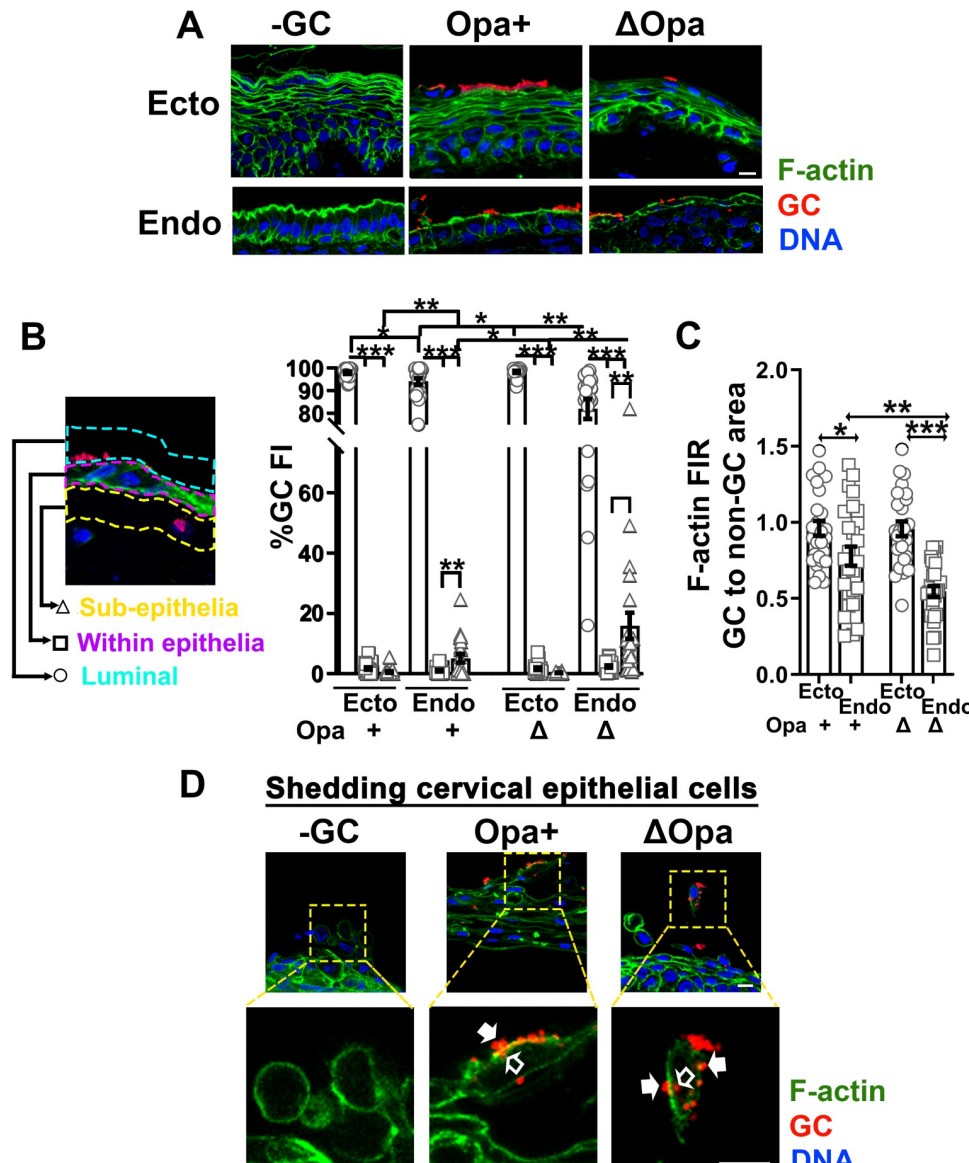

**Fig 6. The level of intraepithelial GC is much lower than those on the luminal surface and/or in the subepithelia of the ecto- and endocervix.** Human cervical tissue explants were incubated with Pil+Opa+ and Pil+ΔOpa (MOI~10) for 24 h, with unassociated GC washed off at 6 and 12 h. Tissue explants were fixed, stained for F-actin, GC and DNA, and analyzed using CFM. **(A)** Shown are representative images of the ecto and endocervix. Scale bar, 10 μm. **(B)** The distribution of GC in cervical tissues was quantified by the percentage of GC fluorescent intensity (% GC FI) detected at the luminal surface (blue dash lines), within the epithelium (magenta dash lines), and at the subepithelium (yellow dash lines), relatively to the total GC FI in the entire area. Data points represent individual randomly acquired images. **(C)** The redistribution of F-actin was quantified by the FIR (±SEM) of phalloidin staining underneath individual GC microcolonies relative to the adjacent no GC luminal area. Data points represent individual GC microcolonies. **(D)** Shown are representative images of shedding cervical epithelial cells stained for GC, F-actin, and DNA. Open arrows, intracellular GC. Filled arrows, surface GC microcolonies with F-actin accumulation. Scale bar, 10 μm. n = 3 three cervixes. *$p<0.05$; **$p<0.01$; ***$p<0.001$.

of ectocervical epithelial cells to GC inoculation, using the tissue explant model and immuno-fluorescence microscopy. In the absence of GC inoculation, we detected ezrin and pMLC staining in the subluminal layers but very little of both in the luminal layer of the ectocervical epithelium (Fig 7A and 7C). We compared the staining levels of ezrin or pMLC between the

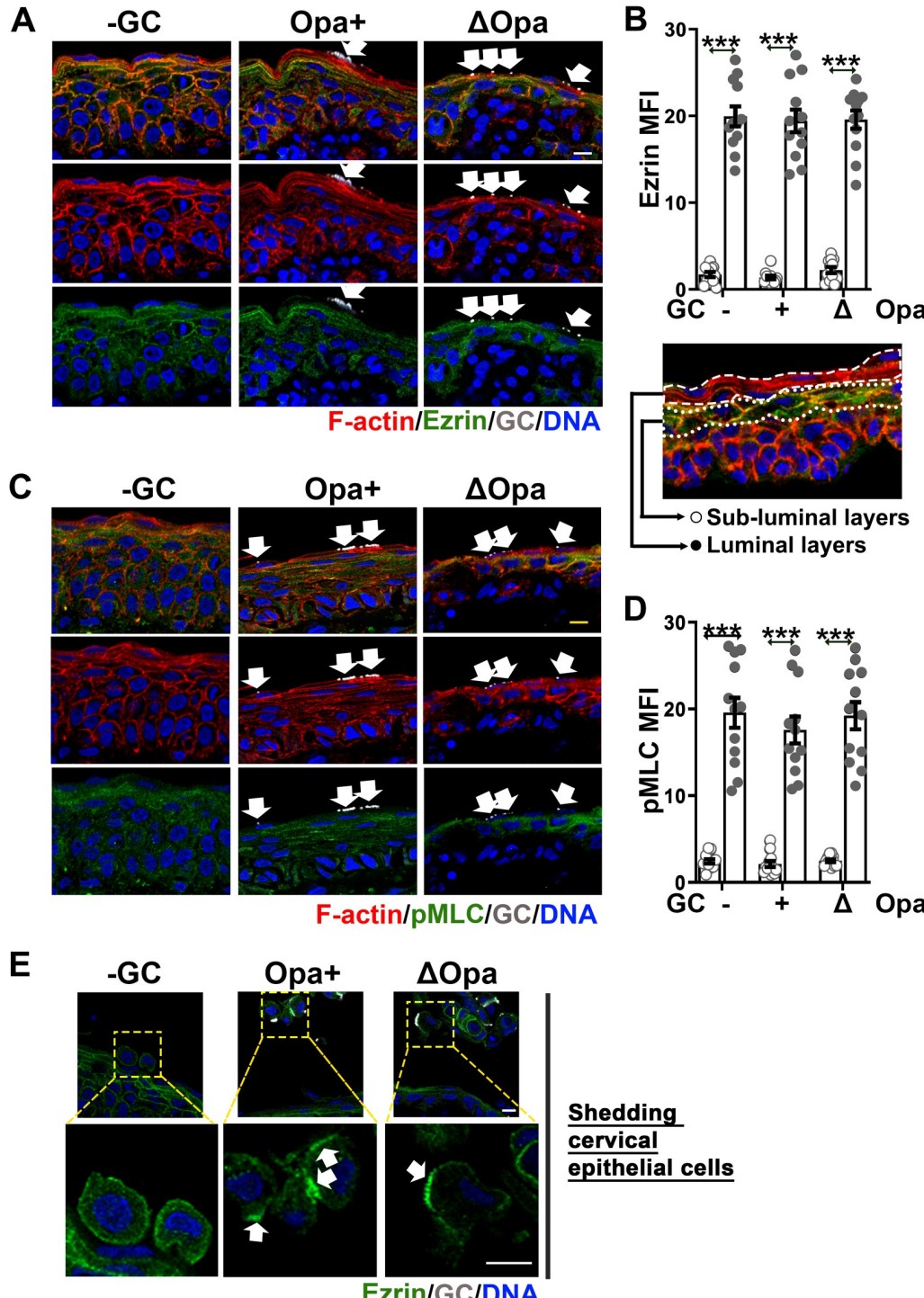

**Fig 7. GC fail to induce ezrin and NMII reorganization in ectocervical epithelial cells.** Human cervical tissue explants were incubated with Pil+Opa+ and Pil+ΔOpa GC (MOI~10) for 24 h and washed at 6 and 12 h to remove unassociated GC. Tissue explants were fixed, stained for ezrin, pMLC, GC and DNA, and analyzed using CFM. **(A, C)** Shown are representative images of ezrin **(A)** and pMLC **(C)** staining in both the luminal and subluminal layers of ectocervical epithelial cells. **(B, D)** The MFI of ezrin **(B)** and pMLC **(D)** in the luminal and the subluminal layers of the ectocervical epithelium. Data points represent individual randomly acquired images. **(E)** Shown are representative images of shedding cervical epithelial cells displaying ezrin accumulation at GC adherent sites (arrows). Scale bar, 10 μm. n = 2 two cervixes. ***$p < 0.001$.

luminal and immediate subluminal layers of the ectocervical epithelium. The MFIs of both ezrin (Fig 7B) and pMLC (Fig 7D) in the luminal layer of epithelial cells were ~10 times less than that in the subluminal layer. Furthermore, GC inoculation did not increase ezrin and pMLC staining at GC adherent sites of the luminal ectocervical epithelial cells (Fig 7A–7D, arrows). However, we detected the accumulation of ezrin staining underneath microcolonies of both Pil+Opa+ and Pil+ΔOpa GC in shedding cervical epithelial cells (Fig 7E, arrows). These data suggest that GC do not induce the recruitment of ezrin and NMII to adherent sites in ectocervical epithelial cells, probably due to their low expression in the luminal layer.

### GC induce the redistribution of Ezrin and NMII in endocervical epithelial cells, leading to F-actin reduction at GC adherent sites

We examined whether GC-induced F-actin reduction in endocervical epithelial cells was caused by ezrin and NMII reorganization, as we observed in the polarized cell line model, using human cervical tissue explants. In the absence of GC, ezrin staining was concentrated at the apical surface of endocervical epithelial cells, colocalizing with apically polarized F-actin (Fig 8A). After GC inoculation, ezrin staining was reduced exclusively underneath Pil+Opa + and Pil+ΔOpa GC microcolonies at the apical surface of the endocervical epithelial cells, where F-actin reduction occurred (Fig 8A, arrows). The FIR of ezrin staining underneath GC microcolonies relative to the adjacent no GC surface area was lower than 1 for both Pil+Opa + and Pil+ΔOpa GC (Fig 8B), suggesting ezrin disassociation from GC adherent sites. The FIR of ezrin staining at the apical surface relatively to the cytoplasm was also significantly reduced, suggesting ezrin disassociation from the apical surface to the cytoplasm (Fig 8C). Both ezrin FIRs, the GC adherent site to the non-GC area and the apical to the cytoplasm, were significantly lower in Pil+ΔOpa GC than Pil+Opa+ GC colonized endocervical epithelial cells (Fig 8B and 8C), suggesting that Opa expression inhibits ezrin redistribution induced by GC in the polarized endocervical epithelial cells. Thus, F-actin reduction at GC adherent sites of endocervical epithelial cells is concurrent with the localized disassociation of ezrin.

We have shown that GC inoculation induces NMII activation and recruitment to GC adherent site at the apical surface of the columnar endocervical epithelial cells [14]. Here, we determined whether such NMII activation led to F-actin reduction at GC adherent sites in endocervical epithelial cells. We pretreated human cervical tissue explants with or without the MLCK inhibitor ML-7 and incubated the explants with Pil+ΔOpa GC in the absence and presence of the inhibitor for 24 h. Tissue sections were analyzed using immunofluorescence microscopy, stained for GC, pMLC for activated NMII, F-actin by phalloidin, and DNA. We observed a recovery of F-actin staining underneath GC microcolonies in endocervical epithelial cells treated with ML-7, compared to those untreated ones (Fig 8D, arrows). The FIR of phalloidin staining at GC adherent sites relative to no GC areas in the endocervical tissues treated with ML-7 (~0.88) was significantly increased compared to those untreated ones (~0.5) (Fig 8E). These data suggest that GC-induced NMII activation is required for the F-actin reduction at GC adherent sites of the columnar endocervical epithelial cells.

## Discussion

While GC invasion into cervical epithelial cells has been well studied using both immortalized and primary cervical epithelial cells [17,21,25,26,32–36,45,78,79], whether GC invade various types of cervical epithelial cells *in vivo* and how the heterogeneity of cervical epithelial cells impacted GC invasion was unknown. Using both cell line and human cervical tissue explant models, this study demonstrates that the polarity of epithelial cells and the expression of ezrin, an actin-plasma membrane linker, are two factors in epithelial cells that regulate GC entry. GC

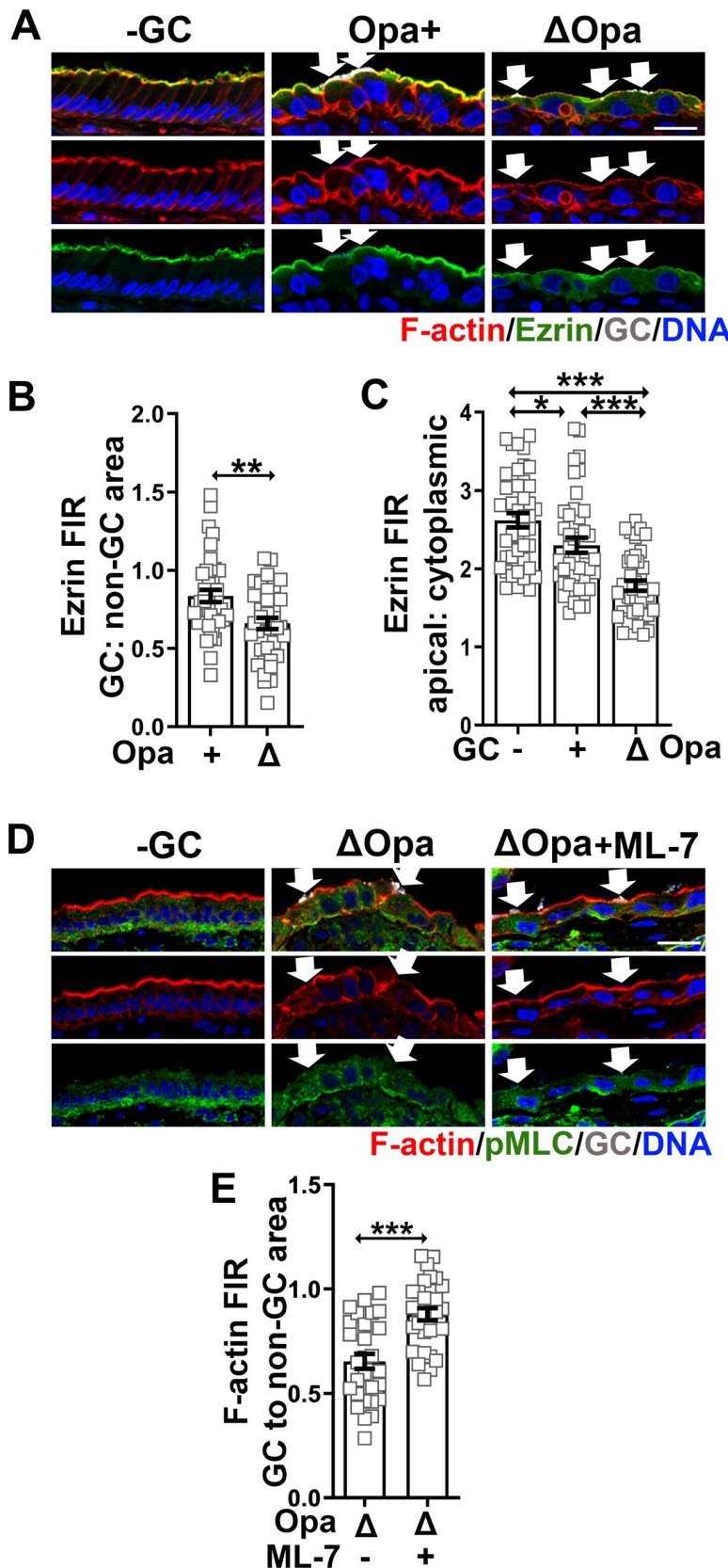

F-actin/Ezrin/GC/DNA

F-actin/pMLC/GC/DNA

**Fig 8. GC inoculation induces ezrin and NMII redistribution in the polarized endocervical epithelial cells, leading to F-actin reduction at GC adherent sites.** Human cervical tissue explants were pretreated with or without ML-7 and incubated with Pil+Opa+ and Pil+ΔOpa GC (MOI~10) for 24 h in the absence or presence of ML-7 (10 μM). Tissue explants were stained for F-actin by phalloidin, ezrin, pMLC, GC and DNA, and analyzed using 3D-CFM. **(A, D)** Shown are representative images of ezrin **(A)** and F-actin **(D)** staining that intercept both the apical and basolateral surfaces of endocervical epithelila cells. Scale bar, 20 μm. **(B, C)** The redistribution of ezrin in the endocervical epithelium was quantified by the FIR (±SEM) of ezrin staining underneath individual GC microcolonies relative to the adjacent no GC surface area **(B)** and by the FIR at the apical surface relatively to the cytoplasm **(C)**. **(E)** The redistribution of F-actin was quantified by the FIR (±SEM) of phalloidin staining underneath individual GC microcolonies relative to the adjacent no GC area. Data points represent individual GC microcolonies. n = 3 three cervixes. $^{**}p < 0.01$; $^{***}p < 0.001$.

invade into non-polarized more efficiently than polarized T84 cells, even though both were derived from the same cell line and express ezrin. GC were detected inside shedding cervical epithelial cells that are not polarized and express ezrin but not much inside non-polarized ectocervical epithelial cells that express little ezrin and polarized endocervical epithelial cells that express ezrin. GC invasion into epithelial cells requires the activation and recruitment of ezrin and ezrin-dependent actin accumulation at GC adherence sites. Reduced GC invasion into polarized epithelial cells is associated with localized decreases in ezrin and F-actin that are normally concentrated at the apical epithelial membrane. Our results support the actin-dependent invasion mechanism shown by many of previous studies [24,26,34,41] and further reveal that GC differentially modulate the actin cytoskeleton and the surface morphology of non-polarized squamous and polarized columnar epithelial cells to facilitate infection.

Early studies have shown that both *Neisseria gonorrhoeae* [34,42–44] and *Neisseria meningitidis* [55,80,81] induce the formation of cortical plaques, consisting of F-actin and ezrin, beneath adherent sites of their targeted host cells, epithelial and endothelial cells, respectively. The actin reorganization is associated with the elongation of host cell membrane protrusions surrounding bacteria, which strengthens bacterial adherence and favors the entry of the bacteria into host cells. Our study has provided additional evidence for an ezrin-dependent GC invasion mechanism. Here, we show that the inhibition of ezrin activation and recruitment by an inhibitor decreases F-actin accumulation at GC adherence sites (Fig 3F and 3G), microvilli elongation (Figs 5A, 5B and S7A), and GC invasion (Fig 3H). While the ezrin inhibitor may have off target effects [82], the low ezrin expression in the luminal layer of the ectocervical epithelium (Fig 7A and 7B) and GC-induced disassociation of ezrin from adherent sites in the apical membrane of columnar endocervical epithelial cells (Fig 3A and 3B) are all associated with reduced GC invasion, supporting the inhibitor data. GC potentially activate ezrin by engaging epithelial surface receptors, such as pili binding to complement receptor 3 (CR3) in collaboration with porin [25,83] and Opa proteins binding to CEACAMs and HSPG [34,39,84,85]. Pili have been shown to be required for the formation of the cortical plaque beneath GC microcolonies in epithelial cells [42,44], and pili-mediated traction forces induce localized accumulation of F-actin and ezrin [53,76]. These interactions trigger signaling and modulate epithelial cell adhesion complexes [78,79,86–88]. Pili, Opa, LOS, and porin have all been shown to promote actin-dependent GC invasion into epithelial cells [24,26,34,41]. Our findings that Opa+ and Opa$_{CEA}$ GC recruit more ezrin and F-actin than those with all *opa* genes deleted and both Opa+ and ΔOpa GC increase ezrin phosphorylation suggest that Opa-host receptor interactions contribute to but not essential for the localized ezrin activation in non-polarized T84 and ME-180 cells. However, how GC differentially activate ezrin in non-polarized and polarized epithelial cells is unclear.

While GC invasion into both non-polarized and polarized epithelial cells has been examined previously [14,21,32,33,36,79,89], this study compared them parallelly using T84 cells

cultured on transwells. Our results reveal that cell polarity is one host factor that can curtail GC invasion into epithelial cells and that GC alter their infection mechanisms based on the polarity of epithelial cells. Even though ezrin and the actin cytoskeleton are highly enriched underneath the apical surface of columnar epithelial cells that GC directly interact with, GC induce sharp reductions rather than further enrichment of ezrin and F-actin and ablation rather than elongation of microvilli at adherent sites of polarized epithelial cells. Such surprising reorganization of ezrin, the actin cytoskeleton, and the apical morphology, opposite to the responses of non-polarized epithelial cells, leads to a reduced GC invasion. Cell polarity is one of the factors distinguishing epithelial cells in the ecto- and endocervix. Evenly distributed cortical actin networks and ezrin in the squamous epithelial cells support a flat cell morphology and randomly appearing microvilli. In contrast, the highly polarized actin cytoskeleton and ezrin in columnar epithelial cells support their vertically extended morphology through the apical junction and dense microvilli at the apical membrane [49]. Such highly concentrated and organized actin networks under the dense apical microvilli are stable and differentially controlled, compared to those under the plasma membrane of non-polarized epithelial cells [49,50]. Furthermore, polarized distribution of epithelial molecules, such as the basally polarized epidermal growth factor receptor (EGFR) and integrins [13,78,90,91], may make ezrin activation signaling spatially unavailable to GC. However, GC induced localized disassembly of ezrin and F-actin networks that support microvilli enables GC to overcome dense and rigid microvilli and expand their interaction with the epithelial cell membrane, securing colonization.

Our previously published data have shown that GC induce $Ca^{2+}$-dependent activation of NMII in both polarized T84 cells and endocervical epithelial cells in tissue explants, which leads to the disassembly of the apical junction, allowing GC to transmigrate across the epithelium [14]. Here, we show that the $Ca^{2+}$-dependent activation of NMII is also a mechanism by which GC trigger the localized disassembly of the actin networks at adherent sites in columnar epithelial cells, as both MLCK and $Ca^{2+}$ flux inhibitors inhibit GC-induced reduction in F-actin (Fig 4E). Even though both MLCK and $Ca^{2+}$ inhibitors may have off-target effects, their consistent effects on GC-induced F-actin reduction support the notion. The disassembly of both the apical junction and ezrin-actin networks at GC adherent sites is likely the result of GC-induced actin reorganization, as perijunctional actomyosin rings and peripheral ezrin-actin networks are essential for the apical junction [92] and microvilli [49], respectively. This study further suggests that the $Ca^{2+}$-dependent NMII activation mechanism appears to be unique to polarized columnar epithelial cells, as activated NMII is recruited to GC adherent sites in polarized T84 cells (Fig 4A and 4B) and endocervical epithelial cells (Fig 8D) [14] but not non-polarized T84 cells (Fig 4A and 4B) and ectocervical epithelial cells (Fig 7C and 7D). Furthermore, we only detected an elevated cytoplasmic $Ca^{2+}$ level in GC-infected polarized but not non-polarized T84 cells 4 h post-inoculation (Fig 4C and 4D). Early studies have shown that GC interaction can trigger a transient $Ca^{2+}$ flux in non-polarized epithelial cells minutes post-inoculation [93–95]. The relationship between GC-induced $Ca^{2+}$ flux at the early time and the elevated cytoplasmic $Ca^{2+}$ level hours later is unclear. Based on the timings, we can speculate that GC may trigger and regulate $Ca^{2+}$ using different signaling mechanisms in non-polarized and polarized epithelial cells, probably due to their distinct organizations of signaling molecules, and the early $Ca^{2+}$ flux may not relate to NMII activation and redistribution.

Consistent with previous reports [27,34,41,66,67], we also found that Opa expression, particularly $Opa_{CEA}$ expression, promotes GC adherence and invasion into both non-polarized and polarized epithelial cells, using gentamicin resistance assay (Fig 1D–1F). In contrast, epithelial cell polarization only reduces GC invasion without affecting adherence (Fig 1A–1C). Our TEM analysis also found a higher percentage of intracellular bacteria in epithelial cells inoculated with Opa+ GC than ΔOpa GC (Fig 1G), indicating that gentamicin-resistance assay

did not overestimate the invasion level of Opa+ GC due to stronger aggregation of Opa+ than ΔOpa GC [68]. Here, we further show that Opa promotes GC adherence and invasion by enhancing ezrin and F-actin accumulation (Figs 2A, 2B, 3A and 3B) and microvilli elongation (Fig 5A and 5B) in non-polarized epithelial cells and inhibiting ezrin-actin network disassembly (Figs 2A, 2B, 3A and 3B) and microvilli modulation (Fig 5C and 5D) in polarized epithelial cells. As 10 out of 11 Opa proteins expressed by MS11 strains bind to CEACAMs [96], Opa-CEACAM interactions likely play the dominant role in the enhanced epithelial actin reorganization in our experimental system. However, our data do not exclude the role of Opa-HSPG interaction in actin-dependent GC invasion when GC predominantly express $Opa_{HSPG}$.

Surprisingly, we did not detect significant GC staining within the ectocervical and endocervical epithelia 24 h post-inoculation, suggesting that GC entery into epithelial cells probably is not a major pathway for GC infection of the human cervix during this time, when compared to colonization and transmigration. However, our observation does not exclude the possibilities of GC invasion into cervical epithelial cells after 24 h and quick exit of epithelial cells after the invasion. The tissue explant model limits our experimental time window and approaches. Even though it is not as sensitive as the gentamicin resistance assay, we clearly visualized intracellular bacteria in epithelial cells shedding from the cervical epithelia using immunofluorescence microscopy, which indicates that this approach is sensitive enough to detect invaded GC. Our observation of intracellular bacteria in shedding cervical epithelial cells is consistent with previous observations using patients' samples [77]. However, whether GC enter cervical epithelial cells before or after shedding is unknown. GC invasion may cause epithelial shedding, and shedding from the epithelium may facilitate GC invasion. GC-induced epithelial exfoliation has been observed in cell lines, mouse, and human cervical tissue explant models [13,14,97,98]. GC can suppress the exfoliation of non-polarized epithelial cells and ectocervical epithelial cells in mice and human tissues through activating integrin, promoting GC colonization [13,97,98]. Meanwhile, GC induce exfoliation of polarized T84 cells and endocervical epithelial cells by activating $Ca^{2+}$ flux and/or NMII, the mechanism that also causes ezrin-actin network and apical junction disassembly, promoting GC transmigration and penetration of cervical epithelia [14]. While this study cannot distinguish these two possibilities, our and previous findings that $Opa_{CEA}$ expression promotes GC invasion but inhibits epithelial shedding [13,14,41,66,98] argue against the possibility that GC entry into epithelial cells before exfoliation.

Comparing the invasion behaviors of GC in T84 cells on transwells and human cervical tissue explants reveals the differences between the two models. Particularly, epithelial cell lines cultured in the non-polarized conditions, which have been widely used as *in vitro* GC infection models, do not form organized multiple layers and differentiate along with layers under the current culture conditions, unlike ectocervical epithelial cells. We have found that the luminal layers of ectocervical epithelial cells reduce the expression of not only ezrin (Fig 7A and 7B) but also β-catenin [13] due to differentiation, which are potential factors regulating GC invasion. However, the GC invasion process in non-polarized T84 and ME-180 cells appears similar to what we observed in shedding cervical epithelial cells and primary cervical epithelial cells isolated from human cervical tissues [45]. T84 cells cultured on transwells for 10 days may not have the same transcriptional and protein expression profiles and polarized distributing molecules exactly as endocervical epithelial cells. On the other hand, the cervical tissue explant model limits experimental approaches. While data interpretation should consider all the limitations, different responses of various epithelial models to GC infection, when being carefully characterized, will help us to better understand the mechanisms underlying GC infection.

This study sheds new light on the mechanisms that GC and human cervical epithelial cells utilize to co-survive in the female reproductive tract. Fig 9 presents a working model for GC

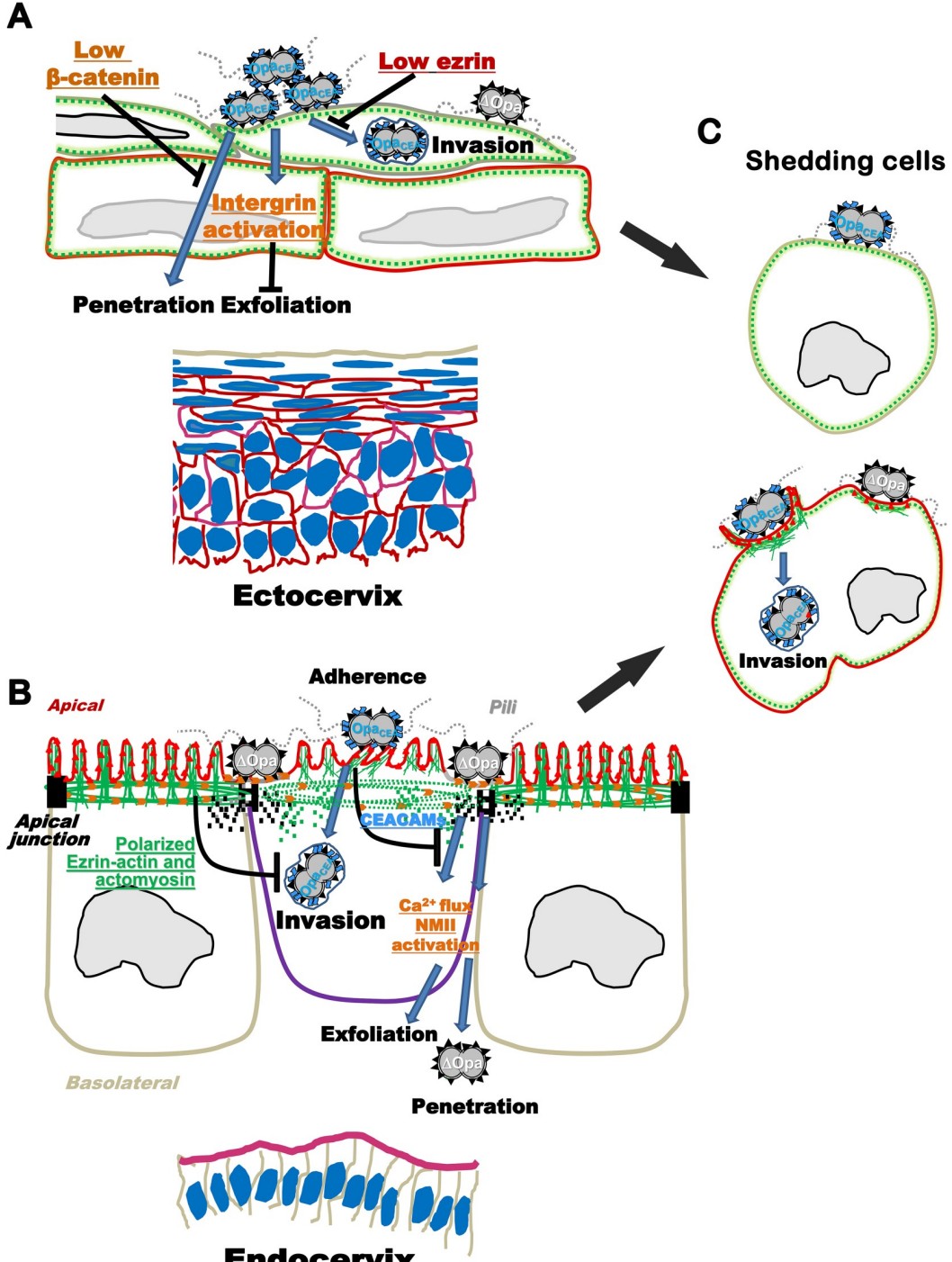

**Fig 9. A working model for GC infection in the human cervix. (A)** GC infection in the ectocervical epithelium. Low expression levels of β-catenin and ezrin in the luminal layer of cervical epithelial cells (A lack of red color at the cell periphery indicates the low expression level of ezrin) potentially inhibit GC-induced actin (green dashed lines at the cell periphery) reorganization. GC also activate β1 integrin in ectocervical epithelial cells, which suppresses epithelial exfoliation and enhances GC adherence. **(B)** GC infection in the endocervical epithelium. GC interactions with the apical surface of columnar cervical epithelial cells with ezrin-actin-supported dense microvilli (ezrin, red triangles; and actin, green lines) activate NMII (orange arrows) at GC adherent sites in a Ca$^{2+}$-dependent manner. The activation and reorganization of NMII induce the disassembly of both ezrin-actin networks underneath GC microcolonies and perijunctional actomyosin rings, which leads to microvilli modification and apical junction disassembly, facilitating GC transmigration across the epithelium and causing epithelial shedding. **(C)** GC infection in ezrin-expressing (red cell border) squamous epithelial cells.

GC interactions activate ezrin (red triangles) and the accumulation of ezrin-actin (green lines) networks at GC adherent sites. This ezrin-actin reorganization drives GC contacting microvilli to elongate and wrap up bacteria, facilitating GC entry into epithelial cells. Pili is essential for initiating GC-epithelial interactions and triggering the events leading to adherence, invasion, and transmigration. Opa proteins, particularly Opa$_{CEA}$, promote GC colonization and invasion by enhancing GC-induced ezrin-actin accumulation and microvilli elongation in squamous epithelial cells **(C)** and inhibiting GC-induced ezrin-actin disassembly and microvilli modification in columnar epithelial cells **(B)**. Opa$_{CEA}$ expression also reduces GC transmigration and columnar epithelial cell shedding by inhibiting GC-induced NMII activation and apical junction disassembly **(B)**.

infection in the human cervix. GC interactions with the luminal surface of the ectocervical epithelium cannot induce efficient invasion and penetration, probably due to the low expression levels of β-catenin [13] and ezrin at the luminal layer of cervical epithelial cells (the low expression level of ezrin is indicated by a lack of red color at the cell periphery) (Fig 9A). GC also activate β1 integrin in ectocervical epithelial cells, which suppresses epithelial exfoliation and enhances GC adherence [13,98] (Fig 9A). GC interactions with the apical surface of columnar cervical epithelial cells with ezrin-actin-supported dense microvilli (Fig 9B; ezrin, red triangles; and actin, green lines) activate NMII [14] (Fig 9B; NMII, orange arrows) at GC adherent sites in a $Ca^{2+}$-dependent manner. The activation and reorganization of NMII induce the disassembly of both ezrin-actin networks underneath GC microcolonies and perijunctional actomyosin rings, which leads to microvilli modification and apical junction disassembly, facilitating GC transmigration across the epithelium and causing epithelial shedding [13,14] (Fig 9B). GC interacting with ezrin-expressing squamous epithelial cells, such as shedding epithelial cells or subluminal layers of ectocervical epithelial cells, activates ezrin and the accumulation of ezrin-actin networks at GC adherent sites (Fig 9C). This ezrin-actin reorganization drives GC contacting microvilli to elongate and wrap up bacteria, facilitating GC entry into epithelial cells (Fig 9C). Pili is essential for initiating GC-epithelial interactions and triggering the events leading to adherence, invasion, and transmigration [13]. Opa proteins, particularly Opa$_{CEA}$, promote GC colonization and invasion by enhancing GC-induced ezrin-actin accumulation and microvilli elongation in squamous epithelial cells (Fig 9C) and inhibiting GC-induced ezrin-actin disassembly and microvilli modification in columnar epithelial cells (Fig 9B). Opa$_{CEA}$ expression also reduces GC transmigration and columnar epithelial cell shedding by inhibiting GC-induced NMII activation and apical junction disassembly [13,14] (Fig 9B). Cervical epithelial cells may protect themselves from GC infection by reducing ezrin and β-catenin expression in the luminal layer of the ectocervical epithelial cells, which associates with low GC invasion and penetration, and by highly polarizing actin-ezrin networks and building rigid microvilli at the apical surface of endocervical epithelial cells, which associate with reduced GC invasion.

## Materials and methods

### Ethics statement

Human cervical tissue explants were obtained through the National Disease Research Interchange (NDRI, Philadelphia, PA). Human cervical tissues used were anonymized. The use of human tissues has been approved by the Institution Review Board of the University of Maryland.

### *Neisseria* strains

*N. gonorrhoeae* strain MS11 that expressed phase variable Pili and Opa (Pil+Opa+), MS11 with all 11 *opa* gene deletion (Pil+ΔOpa), and MS11 Pil+ΔOpa expressing non-phase variable

CEACAM-binding OpaH (Pil+Opa$_{CEA}$) [68] were used. GC were grown on plates with GC media and 1% Kellogg's supplement (GCK) for 16–18 h before inoculation. Pil+ colonies were identified based on colony morphology using a dissecting light microscope. GC was suspended, and the concentration of bacteria was determined by using a spectrophotometer. GC were inoculated at MOI ~10.

## Human epithelial cells

Human colorectal carcinoma cell line T84 cells (ATCC) was maintained in Dulbecco's modified Eagle's medium: Ham F12 (1:1) supplemented with 7% heat-inactivated FBS at 37°C with 5% $CO_2$. To establish polarized epithelium, T84 cells were seeded at 6 x10$^4$ per transwell (6.5 mm diameter, Corning) and cultured for ~10 days until transepithelial electrical resistance (TEER) reached ~2000 Ω. TEER was measured using a Millicell ERS volt-ohm meter (Millipore). To establish non-polarized epithelium, cells were seeded at 2 x10$^5$ per transwell and cultured for 2 days. The 2- and 10-day cultures had the same number of T84 cells per transwell. The human cervical carcinoma cell line ME-180 (ATCC) was maintained in RPMI 1640 media supplemented with 10% heat-inactivated FBS at 37°C with 5% $CO_2$. To examine the adhesion and invasion of GC, ME-180 cells were seeded at 2 x10$^5$ per transwell and cultured for 2 days. To image the response of F-actin and ezrin to GC infection, ME-180 cells were seeded at 4.5 x10$^4$ per transwell and cultured for 2 days to form a monolayer.

## Immunofluorescence analysis of epithelial cells on transwells

Non-polarized and polarized T84 cells or ME-180 cells were pre-treated with or without the ezrin inhibitor NSC668394 (20 μM, EMD Millipore), the myosin light chain kinase (MLCK) inhibitor ML-7 (10 μM, EMD Millipore), or the Ca$^{2+}$ inhibitor 2APB (10 μM, EMD Millipore) for 1 h, and incubated with GC (MOI = 10) for 6 h in the presence or absence of inhibitors. Cells were rinsed, fixed with 4% paraformaldehyde, permeabilized with 0.1% Triton X-100, and stained with phalloidin (Life Technology) and antibodies specific for ezrin (Santa Cruz Biotechnology), pMLC (Cell Signaling Technology) and GC (polyclonal antibodies generated by immunizing goats with the outer membranes of WT MS11 [65]), and Hoechst for nuclei (Life Technologies). Cells were analyzed by confocal fluorescence microscopy (Zeiss LSM 710, Carl Zeiss Microscopy LLC). Z-series of images were obtained in 0.57 μm slices from the top to the bottom of the monolayer, and three-dimensional (3D) composites were obtained using Zen software.

The redistribution of F-actin, ezrin, and pMLC induced by GC inoculation at the luminal surface was examined using xz images across the top to the bottom of epithelia. Fluorescence intensity (FI) profiles along a line underneath each GC microcolony and a line along the adjacent epithelial cell surface without GC were generated using NIH ImageJ software. The fluorescence intensity ratio (FIR) of F-actin, ezrin, and pMLC underneath GC microcolonies relative to the adjacent no GC epithelial surfaces was determined. A total of 54~81 microcolonies from three independent experiments were quantified.

## Human cervical tissue explants

Cervical tissue explants were cultured using a previously published protocol [12]. Briefly, cervical tissues were obtained from patients (28–40 years old) undergoing voluntary hysterectomies through National Disease Research Interchange (NDRI) and received within 24 h post-surgery. Muscle parts of the tissue were removed by using a carbon steel surgical blade. Remaining cervical tissues were cut into ~2.5 cm (L) x 0.6 cm (W) x 0.3 cm (H) pieces, incubated in the medium CMRL-1066 (GIBCO), containing 5% heat-inactivated fetal bovine

serum, 2 mM L-glutamine, bovine insulin (1 μg/ml, Akron Biotech), and penicillin/streptomycin for 24 h and then in antibiotic-free media for another 24 h. Individual pieces of cervical tissue explants were pretreated with or without the MLCK inhibitor ML-7 (10 μM) and inoculated with GC at MOI ~10 (10 bacteria to 1 luminal epithelial cell) in the presence or absence of ML-7. The number of epithelial cells at the luminal surface was determined by the luminal surface area on individual pieces divided by the average luminal area of individual cervical epithelial cells (25 μm$^2$). GC were collected from plates after ~18-h culture and resuspended in pre-warmed antibiotic-free cervical tissue culture media before inoculation. The concentration of bacteria was determined using a spectrophotometer. The cervical tissue explants were incubated with GC at 37˚C with 5% $CO_2$ with gentle shaking for 24 h. The infected cervical tissue explants were rinsed with antibiotic-free cervical tissue culture medium at 6 and 12 h to remove non-adhered bacteria.

### Immunofluorescence analysis of human cervical tissue explants

Tissue explants were fixed by 4% paraformaldehyde 24 h post-inoculation, embedded in 20% gelatin, cryopreserved, and sectioned by cryostat across the luminal and basal surfaces of the epithelium. Tissue sections were stained for F-actin by phalloidin (Cytoskeleton), ezrin (Santa Cruz Biotechnology), pMLC (Cell Signaling Technology), and GC by specific antibodies, and nuclei by Hoechst (Life Technologies), and analyzed using a confocal fluorescence microscope (Zeiss LSM 710, Carl Zeiss Microscopy LLC) as previously described [12]. Images of epithelia were randomly acquired as single images or Z-series of 0.57 μm/image, and 3D composites were generated using Zeiss Zen software.

GC distribution in cervical tissue explants was measured by the percentage of GC fluorescence intensity (% GC FI) at the luminal surface, within the cervical epithelium, and at the subepithelium. Seven randomly acquired images of each cervical region from each of three cervixes were quantified. The relative staining levels of ezrin or pMLC in the luminal layers and sub-luminal layers of ectocervical epithelial cells were evaluated by mean fluorescence intensity (MFI) in individual images. Six randomly acquired images from each of two cervixes were quantified. The redistribution of F-actin and ezrin in endocervical epithelial cells was evaluated by the fluorescence intensity ratio (FIR) underneath GC microcolonies relative to the adjacent surface areas without GC attached, or at the apical surface relative to in the cytoplasm in individual endocervical epithelial cells. A total of 27~52 microcolonies from images of three cervixes were quantified.

### GC adherence and invasion assays

The adherence and invasion of GC in T84 and ME-180 cells were analyzed as previously described [26,65]. Briefly, non-polarized and polarized epithelial cells were pretreated with or without ezrin inhibitor, NSC668394 (20 μM), for 1 h, inoculated with GC in the presence or absence of the inhibitor from the top chamber for 3, 6, and 12 h to analyze adherence and invasion. For adherence analysis, GC-inoculated cells were washed intensively, lysed, and plated on GCK plates to determine the CFU. For invasion analysis, GC-inoculated cells were treated with gentamicin (100 μg/ml) for 2 h, washed, lysed, and plated on GCK plates to numerate CFU. Gentamicin-resistant bacteria were counted as invaded GC. Three transwells from each of three independent experiments were quantified.

### Calcium imaging

Non-polarized and polarized T84 cells on transwells were pretreated with or without thapsigargin (10 μM, Sigma) for 1 h and incubated with GC (MOI~10) apically with or without the

inhibitors for 4 h. Thapsigargin is an endoplasmic reticulum $Ca^{2+}$ ATPase inhibitor that causes $Ca^{2+}$ elevation in the cytoplasm and thereby served as a positive control. Cells were incubated with the fluorescent calcium indicator Fluo-4 AM ester (100 μM, Life Technologies) for 1 h, and xz images were acquired in the presence of the membrane dye CellMask (5 μg/ml, Life Technology) using Leica TCS SP5X confocal microscope (Leica Microsystems). The MFI of Fluo-4 in the cytoplasmic region in individual cells was measured using the NIH ImageJ software. Twenty randomly selected cells from each of three independent experiments were quantified.

## Function analysis of the apical junction

Polarized and non-polarized T84 cells were incubated with the CellMask dye (5 μg/ml, Life Technologies) in the bottom chamber for 15 min, and xz images were acquired using Leica TCS SP5 X confocal microscope (Leica Microsystems). The percentage of epithelial cells displaying CellMask staining at the apical membrane in each randomly acquired image was determined.

## Western blotting

Non-polarized and polarized T84 cells were pretreated with or without the ezrin inhibitor NSC668394 (20 μM) for 1 h and incubated with GC with or without the inhibitor from the top chamber for 6 h, and then lysed by RIPA buffer [0.1% Triton x100, 0.5% deoxycholate, 0.1% SDS, 50 mM Tris-HCl, pH 7.4, 150 mM NaCl, 1 mM EGTA, 2 mM EDTA, 1 mM $Na_3VO_4$, 50 mM NaF, 10 mM $Na_2PO_4$, and proteinase inhibitor cocktail (Sigma-Aldrich, St. Louis, MO)]. Lysates were resolved using SDS-PAGE gels (BioRad) and analyzed by Western blot. Blots were stained for ezrin (Cell Signaling Technology) and phosphorylated ezrin at T567 (Abcam), stripped and reprobed with anti-β-tubulin antibody (Sigma). Blots were imaged using a Fujifilm LAS-3000 (Fujifilm Medical Systems). Each data point represents individual transwells. Two transwells from two to three independent experiments were quantified.

## Transmission electron microscopic analysis

Non-polarized and polarized T84 cells were pretreated with or without the ezrin inhibitor NSC668394 (20 μM) for 1 h, and incubated with GC with or without the inhibitor from the top chamber for 6 h with MOI ~50. The epithelial cells on the transwell membrane were fixed, processed, and embedded using EPON 812 resin (Araldite/Medcast; Ted Pella). Thin sections crossing the top and the bottom membranes were collected, stained with uranyl acetate and lead citrate, and imaged by a ZEISS 10CA electron microscope (ZEISS). Microvilli elongation and ablation at the luminal membrane were quantified using 3~4 randomly acquired images at a magnification of 5k from each of 3 independent experiments. The percentage of intracellular GC was estimated using two out of eight randomly acquired images showing intracellular GC over the total number of GC directly contacting the apical membrane at a magnification of 5k from each of three independent experiments.

## Statistical analysis

Statistical significance was assessed by using Student's *t*-test by Prism software (GraphPad). P-values were determined using unpaired *t*-test with Welch's correction in comparison with no infection controls.

## Supporting information

**S1 Fig. The barrier function of non-polarized and polarized T84 epithelial cell line.** T84 cells were cultured on transwells for 2 or 10 days. **(A)** Transepithelial electric resistance (TEER) indicating the ion permeability of the epithelium. **(B-D)** The 2- and 10-day T84 cells were cultured with the CellMask lipid dye in the bottom chamber for 15 min and imaged using a confocal fluorescence microscope (CFM). Shown are representative xz images **(B)**, the percentage of cells with (black) and without (white) CellMask dye diffusion from the bottom into the top chamber **(C)**, and the fluorescence intensity ratio (FIR) of CellMask staining at the top to the bottom membrane of T84 cells **(D)**. n = 3 three independent experiments. ***$p<0.001$, Scale bar, 10 μm. (TIF)

**S2 Fig. Distribution of F-actin and junctional proteins in the non-polarized and polarized T84 epithelial cell and the cervical tissue explant models. (A-C)** T84 cells were cultured on transwells for 2 or 10 days, stained for E-cadherin **(A)**, ZO-1 **(B)**, F-actin **(C)**, and Hoechst, and imaged by a CFM. Shown are representative xz images. **(D-F)** Human cervical tissue explants were cultured for two days, fixed, and cryopreserved, sectioned, stained for E-cadherin **(D)**, ZO-1 **(E)**, F-actin **(F)**, and Hoechst, and imaged using a CFM. Shown are representative images across epithelia. n = 4 four independent experiments. Scale bar, 10 μm. (TIF)

**S3 Fig. Opa expression enhances both GC adherence and invasion into non-polarized ME-180 cells.** ME-180 were cultured on transwells for 2 days and pretreated with or without the ezrin activation inhibitor NSC668394 (20 μM) for 1 h and inoculated with Pil+Opa+, Opa$_{CEA}$ or ΔOpa GC (MOI~10) from the top for 3 or 6 h with or without the inhibitor. Infected epithelial cells were lysed and cultured before and after gentamicin treatment to determine total epithelial-associated GC (±SEM) at 3 h **(A)** and gentamicin-resistant GC (±SEM) at 6 h **(B)**. Data points represent individual transwells. n = 2 two independent experiments and three transwells per experiment. *$p<0.05$; **$p< 0.01$; ***$p<0.001$. (TIF)

**S4 Fig. GC invade into non-polarized epithelial cells.** Non-polarized and polarized T84 cells were incubated with Pil+Opa+ or ΔOpa GC (MOI~50) from the top of the chamber for 6 h, fixed, and processed for transmission electron microscopy (TEM). Shown are three sets of images of non-polarized T84 cells with intracellular Opa+ (left) or ΔOpa GC (right). In each set of images, the right panels are original images, and the left panels are enlarged areas within the yellow dash lines. Small arrows, GC directly contacting epithelial membranes. Big arrows, intracellular membrane. (TIF)

**S5 Fig. Opa expression enhances the F-actin and ezrin enrichment in non-polarized ME-180 cells.** ME-180 were cultured on transwells for 2 days and pretreated with or without the ezrin activation inhibitor NSC668394 (20 μM) for 1 h and inoculated with Pil+Opa+, Opa$_{CEA}$ or ΔOpa GC (MOI~10) from the top of transwells for 6 h with or without the inhibitor. Cells were fixed, stained for F-actin, ezrin, and GC, and analyzed using 3D-CFM. Representative xy images of the top surface and xz images crossing the top and the bottom surfaces are shown **(A, C)**. Arrows indicate the location of GC. The redistribution of F-actin **(B)** and ezrin **(D)** was quantified by the mean fluorescence intensity ratio (FIR) (±SEM) of F-actin underneath individual GC microcolonies relative to the adjacent no GC surface area. Data points represent individual GC microcolonies. Scale bar, 20 μm. n = 2 two independent experiments. *$p<0.05$;

***$p < 0.001$.
(TIF)

**S6 Fig. The ezrin inhibitor does not affect GC growth and adhesion. (A)** MS11 Pil+Opa
+ GC were cultured in DMEM/F12 containing 10% FBS for 6 h in the absence or presence of
the ezrin inhibitor NSC668394 (20 μM). CFU was determined by serial dilution and plating on
GCK plates. Data points represent individual wells. n = 2 two independent experiments and
three wells per experiment. **(B)** Non-polarized T84 cells were pretreated with or without the
ezrin activation inhibitor NSC668394 (20 μM) for 1 h and inoculated with Pil+Opa+ GC
(MOI~10) from the top of transwells for 3 h with or without the inhibitor. Total T84-associated GC (±SEM) were quantified by culturing the lysates of infected epithelial cells. n = 2 two
independent experiments and three transwells per experiment.
(TIF)

**S7 Fig. GC differentially remodel the surface morphology of non-polarized and polarized
epithelial cells in an ezrin- or NMII-dependent manner.** Non-polarized and polarized T84
cells were pre-treated with or without NSC668394 (20 μM) or ML-7 (10 μM) and incubated
with Pil+Opa+ or Pil+ΔOpa GC (MOI~50) from the top of transwells for 6 h with or without
the inhibitors. Cells were fixed and processed for TEM. Shown are two sets of example images
with high (left panels) and low (right panels) magnifications of non-polarized **(A)** and polarized **(B)** T84 cells. Yellow dash line rectangles highlight the focused area. Filled arrows, GC-
associated elongated microvilli. Open arrows, GC contacting bending microvilli. Open arrow-
heads, no microvilli visible at GC contact sites. Scale bar, 1 μm.
(TIF)

**S8 Fig. Intracellular GC and F-actin accumulation at adherent sites in epithelial cells shed-
ding from the cervical explant tissue.** Human cervical tissue explants were incubated with Pil
+Opa+ and Pil+ΔOpa (MOI~10) for 24 h, with unassociated GC washed off at 6 and 12 h. Tis-
sue explants were fixed, stained for F-actin, DNA and GC, and analyzed using CFM. Two sets
of example images show intracellular GC **(A)** and F-actin accumulation at GC adherent sites
**(B)** in shedding cervical epithelial cells. Open arrows, intracellular GC. Filled arrows, surface
GC microcolonies with F-actin accumulation. Scale bar, 10 μm.
(TIF)

## Acknowledgments

We thank the UMD CBMG Imaging Core for all microscopy experiments and the Laboratory
for Biological Ultrastructure for all the transmission electron microscope experiments.

## Author Contributions

**Conceptualization:** Qian Yu, Liang-Chun Wang, Wenxia Song.

**Data curation:** Qian Yu, Liang-Chun Wang, Sofia Di Benigno.

**Formal analysis:** Qian Yu, Liang-Chun Wang.

**Funding acquisition:** Daniel C. Stein, Wenxia Song.

**Investigation:** Qian Yu, Liang-Chun Wang.

**Methodology:** Qian Yu, Liang-Chun Wang.

**Project administration:** Wenxia Song.

**Resources:** Daniel C. Stein.

**Supervision:** Wenxia Song.

**Validation:** Qian Yu.

**Visualization:** Qian Yu, Liang-Chun Wang.

**Writing – original draft:** Qian Yu.

**Writing – review & editing:** Liang-Chun Wang, Daniel C. Stein, Wenxia Song.

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
