## [Decision Letter · Decision Letter 0]

14 Jun 2021

Dear Dr. Song,

Thank you very much for submitting your manuscript "Gonococcal invasion into epithelial cells depends on both cell polarity and ezrin" (PPATHOGENS-D-21-00895) for consideration at PLOS Pathogens. As with all papers peer reviewed by the journal, your manuscript was reviewed by members of the editorial board and by several independent peer reviewers. Based on the reports, we regret to inform you that we will not be pursuing this manuscript for publication at PLOS Pathogens.

The reviewers and I all agree there is merit in the work regarding the importance of epithelial polarity in gonococcal interactions with epithelial cells, in particular the role of ezrin in cytoskeletal dynamics that are important for invasion. However, many points were raised in review, some by more than one reviewer, that prevent acceptance of the manuscript without substantial editing and additional experimentation outside the time frame of the usual revision period. In particular, please consider 1) analyzing more than one time point in binding and invasion, 2) using GC constitutively expressing a single Opa in the isogenic delta Opa background for Opa-dependent effects, 3) considering that exfoliation may occur as a consequence, not cause of invasion, 4) rewriting to frame in light of prior literature on this topic, 5) addressing the discrepancy between gentamicin and TEM data, and addressing the biological significance of the small but statistically significant differences reported, and 6) using less strong language to describe the results and their interpretation.

The reviews are attached below this email, and we hope you will find them helpful if you decide to revise the manuscript for submission elsewhere. We are sorry that we cannot be more positive on this occasion. We very much appreciate your wish to present your work in one of PLOS's Open Access publications. 

Thank you for your support, and we hope that you will consider PLOS Pathogens for other submissions in the future.

Sincerely,

Alison Criss

Guest Editor

PLOS Pathogens

Christoph Tang

Section Editor

PLOS Pathogens

Kasturi Haldar

Editor-in-Chief

PLOS Pathogens

orcid.org/0000-0001-5065-158X

Michael Malim

Editor-in-Chief

PLOS Pathogens

orcid.org/0000-0002-7699-2064

Reviewer's Responses to Questions

**Part I - Summary**

Reviewer #1: This work examines the differences in the capacity of GC to invade polarized and nonpolarized epithelial cells using a polarizable cell line (T84 cells, intestinal) and cervical explants, which contain the different epithelial cell types within the cervix. Piliated, Opa-positive and piliated Opa-negative gonococci are compared throughout the study. Strengths of the manuscript include i.) the state-of-the art microscopy; ii.) the use of a highly relevant cell culture system (cervical explants) that reproduces the heterogeneity of the histology of the cervix, and iii.) investigation of the mechanism of invasion from the viewpoint of the host cell, specifically whether actin polymerization, ezrin recuritment and phosphorylation differ among these cell types and in response to GC. The mechanistic studies were nicely hypothesis-driven. 

Areas where this work could be improved by i.) more discussion of why nonpolarized T84 cells are likely irrelevant system for studying GC invasion, based on what was found with ectocervical cells in cervical explants (which don’t have polarity), ii.) adding experimentation with nonpolarized cervical epithelial cells used by many other laboratories to determine whether these are also likely not representing events that occur in whole model systems (i.e. explants) (i.e. it would be very interesting to know whether actin polymerization, ezrin recruitment, etc. occurs in response to GC in ME180 cells in a way that is similar to that of nonpolarized T84 cells), and iii.) the lack of staining polarized and nonpolarized cells for CEACAMs, which seems important in light of occasional differences between Opa-positive versus Opa-negative bacteria, and the lack of genetic complementation for in experiments in which a difference between Opa-positive and Opa-negative GC was found. 

Furthermore, the authors show the interesting observation that intracellular GC are seen in shed cervical epithelial cells but do not discuss this information in the context of Hauck paper (Muenzner et al J Cell Biol. 2005) which demonstrates the effect of Opa engagement of CEACAMS on epithelial cell shedding. (there are several other papers or reviews by this group that discuss this topic).

It would also be useful to the reader to place these results, as well as the rationale of the study, in the context of what was previously reported by these authors using the cervical explant system (Yu et al, PLoSPath 2013)so that a larger view of GC invasion through and between cells, with and without CEACAMs could be made. A figure that depicts a model that ties these two pieces of work together would be very useful in summarizing what the cervical explant system has shown.

Reviewer #2: In this study, Yu et al. employ two in vitro models (immortalized T84 cells, and cervical explants) to analyze the mechanisms of gonococcal invasion in cells with various morphologies. Building on prior work from their group and others, they present descriptive results on cell morphology during infection, as well as perturbation experiments which suggest roles for actin, the actin-membrane linker ezrin, and signaling events including ezrin phosphorylation and Ca2+ signaling in host cell responses and bacterial invasion. 

In general the experiments are well-controlled and described and the experimentation is extensive. The major experimental limitations (see below) are the use of pharmacological inhibitors that may have off-target effects, and a focus on analyses conducted at only a single time point (in most cases, 6 h post-infection). Most of the microscopy is beautiful.

The manuscript itself raises more significant concerns. As mentioned above, the experiments build on older work. Key papers, sometimes with nearly identical experiments, are not cited as prior art, and in some cases the data presented appear at odds with older results. It will be essential in any resubmission to address these issues. In particular, divergence between the present manuscript and other published data should be highlighted so that inconsistencies can be addressed (and, one hopes, resolved) in future work.

Reviewer #3: Yu et al. examine use microscopy to examine the roles of cell polarization, ezrin, and F-actin in gonococcal interactions with human cells. An invasion method is described that occurred in non-polarized T84 colonic cells and in cervical cells found in the medium of cervical explant cultures, though the mechanisms leading to cell infection and exfoliation were not determined. By contrast, invasion was said not to occur in polarized T84 cells or in cervical explant tissue, raising questions about the significance of this invasion phenomenon.

**Part II – Major Issues: Key Experiments Required for Acceptance**

Reviewer #1: 1. Experiments should be done with nonpolarized cervical cells to determine whether the high rate of invasion of nonpolarized T84 cells and changes in actin, ezrin and phosphorylation are unique to the T84 intestinal cell line, and to confirm this important and interesting observation in a frequently used cervical cell line.

2, A complemented opa-negative mutant should be used for all experiments in which a difference was found for Pil+,Opa+ versus Pil+, Opa- GC. Introduction of a single opa gene should be sufficient for the complemented strain. CEACAM staining should also be incorporated into the immunofluorescence studies.

Reviewer #2: Most of the following issues do NOT require additional experimentation, but are still important. It might be possible, with extensive rewriting, or presentation of control data collected but not presented, to address the following concerns without additional experimentation. 

1. The authors seem to imply (Results, first paragraph) that the choice of the T84 model, or its use in polarized and non-polarized configurations, is novel. It is not. The first use of this system to study both N. meningitidis and N. gonorrhoeae infection was published in 1996 (Merz et al., PMID: 8972489). Reassuringly, the adhesion, invasion, and (for polarized cells) transepithelial migration data presented here are very close to the data that we reported for both polarized and non-polarized T84 cells 25 years ago. Many of the T84 results are also consistent with data published soon thereafter by others who are *also* not cited (Criss et al., PMID: 16922862; Pujol et al., Wang et al., PMID: 17683982; Pujol et al., PMID: 9353073). These papers need to be cited.

1a. Similarly, on p. 22, the authors write: "Our findings that Opa expression promotes GC invasion into T84 epithelial cells, no matter if they are non-polarized and polarized using gentamicin resistant assay, are consistent with previous reports [45]." But that paper (Grassmé) does not employ T84 or any other polarized cell line. It uses Chang cells, now known to be a HeLa derivative. 

1b. In Fig. 1, why are the invasion indices (ratio of invasion : cell association) so much higher when assessed by TEM vs. Gm protection assay?

2. The authors say (P. 12, 1st paragraph) that Ca2+ signaling does not occur with non-polarized cells (Fig. 3). That claim appears to be in direct contradiction to prior results (again, key work by other authors is not cited — e.g., Källström et al., PMID: 9705315; Ayala et al., PMID: 11298650 and PMID: 16309460; Müller et al., PMID: 9889191). However, the earlier experiments demonstrated Ca2+ signaling on a time scale of minutes, while the present manuscript apparently assays elevated Ca2+ only at a 4h time point. 

2a. The prior art needs to be cited and discrepancies between those studies and the present work need to at least be mentioned and preferably discussed. 

2b. Given previously reported results, what might have happened during the previous 4h?

2c. Minor point: the authors say that Fluo-4 is used to detect Ca2+, but Fluo-4 is cell-impermeant. If an AM ester was used that should be specified and the product number noted. 

3. The descriptions of the roles of microvilli are very interesting, but could be considerably clarified and sharpened. For example, it's pretty clear that it's microvillus *remodeling or dynamics*, not the presence or absence of microvilli per se, that controls the ability of the pathogen to invade cells. That's clearly stated in the Discussion (top of p. 20), but much less clear elsewhere in the manuscript. 

4. The general conclusion that polarity, per se, (and especially cytoskeleton polarity) is controlling GC entry, is found throughout the manuscript, but is not particularly well-supported. The development of polarity is accompanied by other changes that occur during differentiation, such as the sequestration of receptors on the basolateral surface or differences in expression of relevant receptor or downstream signaling molecules. Modulating polarity by simply changing the time after plating the cells onto transwell filters does not control for these potential differences. This limitation needs to be discussed. 

5. The small molecule inhibitors used, particularly the ezrin inhibitor, are not particularly well-characterized. 2-APB is known to have fairly complex effects and to differentially target different classes of Ca2+ channels, even acting as an *agonsist* of Trpv channels at concentrations similar to those used here (Gao, PMID: 26876731) Again, the experimental limitation should at least be mentioned, with appropriate pointers to literature. 

6. The experiments with cervical explants are particularly useful, and represent a real step forward. However, there are again some unstated limitations. 

6a. First and foremost, there is the curious result that little or no invasion was detected in endocervical or ectocervical cells. Instead, bacteria were noted in shed cells, leading to the authors' conclusion that GC may prefer to enter shedding cells. The alternative conclusion, of course, is that GC entry *causes* the cells to delaminate. Both alternative hypotheses should be mentioned. 

6b. The lack of invasion noted in explants could conceivably reflect not just terminal differentiation or cornification, but cell death. Are the cells that are not invaded alive? Was viability examined?

7. As mentioned above, the study's biggest limitation, to my mind, is the examination of single time points rather than trajectories. The choices of time points need to be explained, and the limitations of using single time points (we can't see what we don't look at) need to be forthrightly discussed.

Reviewer #3: Major points.

1. The biological importance of the differences in FIR reported throughout the paper are difficult to discern, particularly when the differences are so small. For instance, in Fig. 1F the authors report a statistically-significant difference in F-actin FIR between Opa+ GC infections and delta-opa GC infections of non-polarized cells, and yet the averages appear to be 1.4 and 1.2, respectively, and the points in the delta-opa infection condition appear to completely overlap with those in the Opa+ condition, with only eight points in the Opa+ condition outside this range. Are these results actually different? Certainly the micrographs provided (Fig 1E) do not show any discernable difference between the Opa+ and delta-opa infections. 

2. Fig. 1 claims to report both invasion (1B) and intracellular GC (1D), yet these two experiments yield different results. The authors conclude from 1D that no intracellular GC are present in polarized cells, while the standard invasion assay in 1B shows maybe 30% as many GC in the polarized cells as in the non-polarized. 

3. As in Fig 1, the results for Opa+ and delta-opa in Fig 2B and 2G appear very small. The error bars presented do not appear to represent the large spread in the data as represented by the data points shown. 

4. It is unclear if the microscopy experiments were repeated, i.e., does the “n=3” found in the figure legends mean that the experiment was done on three different days with three different cultures, or does it mean that it was done once with three technical replicates?

5. One of the most interesting results of the study is the observation of sloughed cervical cells with internalized gonococci. The authors claim both that quantification of this process is impossible and that the number of gonococci that invaded cervical cells is less than the number that remain on the surface. Isn’t it possible that gonococci invaded significant numbers of cervical cells that were then sloughed from the epithelium? They should collect sloughed cells from the medium and plate the bacteria, add more earlier time points to the microscopy studies to try to see the invasion and sloughing process, and quantify cell sloughing as has been done by Muenzner (Science 329:1197). 

6. The authors’ previous cervical tissue studies identified the transition zone as the region of the cervix where gonococci invade tissue (Yu et al. doi.org/10.1371/journal.ppat.1008136). Since the authors in that study reported non-polarized cells in the transition zone, it seems quite possible that the invasion phenomenon studied in the current manuscript could be going on in those cells. Why was the transition zone not examined in this study? Thus the conclusions in the last paragraph of the discussion that the cervix is protected from GC invasion may be wrong. 

7. The authors should explain why their results with T84 cells are different from those of Merz and So, particularly with regard to invasion and transcytosis (doi: 10.1146/annurev.cellbio.16.1.423.).

**Part III – Minor Issues: Editorial and Data Presentation Modifications**

Reviewer #1: The lack of line numbers makes reviewing this manuscript difficult.

Experimental:

Data pertaining to Fig. 1: A brief description of the details of this experiment with respect to time points should be presented before discussing the data. Adherence was measured at 3 hrs; gentamicin-protected GC were measured at 6 hrs. When was the gentamicin added? While this information is in the methods, it would help the reader understand the data better if the assay is described better here. Similarly, to describe the T84 cells used as cells incubated for 2 days or 10 days in many figure legends is not as informative as describing them as nonpolarized and polarized as was done in the Figure 3 legend.

Differences in the results when based on the Gm protecton assay: While the greater sensitivity of the Gm protection assay compared to microscopy methods is discussed, it is also possible that the Gm protection assay if incomplete exposure of extracellular GC to the Gm due to micro-crevices/folding in the cells or inadequate washing. This finding is also potentially useful to others who use this well-established assay. The recovery of greater than 1,000 Gm-protected CFU, for example, in Figure 1 corresponds to a value of 16 % or 7% intracellular GC (number of GC/100 epithelial cells). Is this difference in the two assays consistent for all experiments done in this paper? 

Occasionally there is a difference when using Opa-positive versus Opa-negative Gc, for example in the enrichment of ezrin in infected nonpolarized cells and the detection of intracellular GC in endocervical and ectocervical cells (Fig. 5A). A complemented Opa-negative strain (i.e. with one opa gene) should be used to confirm these differences are due to lack of Opa expression. 

With respect to differences in results when Opa-positive GC are used, CEACAM staining should be performed on polarized and nonpolarized T84 cells to confirm that Opa interactions with these receptors is occurring (or not), and to see if the CEACAMs are on the apical side of polarized cells or localized (i.e. recruited) differently when GC are present.

Editorial:

There are many studies on GC invasion of nonpolarized ME180 cells and primary cervical cells. This should be discussed with respect to the potential lack of polarity in these specific cell systems and how that may influence actin and ezrin recruitment and localization. Could it be that GC is not as invasive as is thought, based on these nonpolarized cell models? Also, no mention is made of the well-defined invasion pathway discovered by Jennifer Edwards, which leaves the discussion incomplete for the reader who is trying to put this all together.

The cervical explant system used by the authors is a unique and powerful system in its relevance and capacity to compare differences in different anatomical sites within the cervix. It is puzzling to this reviewer why they do not build on their previous work (PLosPath 2015) with this system to better explain what they are testing and why (in the introduction) and to provide the reviewer with a larger and more comprehensive summary of what has been learned with this system in the discussion. 

Shedding should be discussed in the context of C. Hauck's work in this area.

Minor:

Last sentence of top paragraph, page 10, is incomplete or missing some words.

The opening line of the abstract states that GC causes symptomatic infections in the cervix but in the introduction, more emphasis is placed on the high percentage of asymptomatic infections in women. This is confusing and because symptomology doesn’t have much to do with this paper, this could be removed from the abstract.

Reviewer #2: (No Response)

Reviewer #3: Minor points.

1. The use of red and green for microscopy images is not optimal for color-blind readers. 

2. The addition of more micrographs (added to the supplement) or larger micrographs might better support the study.

3. Methodological details are missing. What is the anti-gonococcal antibody used? What method was used for permeabilization of cells? 

4. The labeling is wrong in Fig 1D (assuming the text correctly describes the results). 

5. P3, line 2. Note that gonorrhea is the second most common reportable STI, not the second most common STI. 

6. P4, line 11. “disassembly”

7. P6, line 16. Which CEACAMs are on T84s?

8. P10, line 7. Awkward, reword.

9. Fig 3F label. F-actin

10. P15, line 3. Only one strain of GC was used in the study. Reword to clarify to WT and mutant.

11. P18, last line. “Prefer” is too much anthropomorphizing. Reword. 

12. Include some figure references in the discussion to lead the reader to the points being made. 

13. P20, line 4. “Pili have”.

PLOS authors have the option to publish the peer review history of their article (what does this mean?). If published, this will include your full peer review and any attached files.

Reviewer #1: No

Reviewer #2: Yes: Alexey Merz

Reviewer #3: No

---

## [Editor Report · Decision Letter 1]

15 Nov 2021

Dear Dr. Song,

We are pleased to inform you that your manuscript 'Gonococcal invasion into epithelial cells depends on both cell polarity and ezrin' has been provisionally accepted for publication in PLOS Pathogens. Your edits have substantially improved the manuscript, which contains interesting novel data and acknowledges the relevant prior literature.

Best regards,

Alison K Criss

Guest Editor

PLOS Pathogens

Christoph Tang

Section Editor

PLOS Pathogens

Kasturi Haldar

Editor-in-Chief

PLOS Pathogens

orcid.org/0000-0001-5065-158X

Michael Malim

Editor-in-Chief

PLOS Pathogens

orcid.org/0000-0002-7699-2064
---

## [Editor Report · Acceptance letter]

24 Nov 2021

Dear Dr. Song,

We are delighted to inform you that your manuscript, "Gonococcal invasion into epithelial cells depends on both cell polarity and ezrin," has been formally accepted for publication in PLOS Pathogens.

Best regards,

Kasturi Haldar

Editor-in-Chief

PLOS Pathogens

orcid.org/0000-0001-5065-158X

Michael Malim

Editor-in-Chief

PLOS Pathogens

orcid.org/0000-0002-7699-2064